# Assessing the Efficacy of Whole-Body Titanium Dental Implant Surface Modifications in Inducing Adhesion, Proliferation, and Osteogenesis in Human Adipose Tissue Stem Cells

**DOI:** 10.3390/jfb13040206

**Published:** 2022-10-27

**Authors:** Federico Ferro, Federico Azzolin, Renza Spelat, Lorenzo Bevilacqua, Michele Maglione

**Affiliations:** 1Department of Medical and Biological Sciences, University of Udine, 33100 Udine, Italy; 2Department of Medical, Surgery and Health Sciences, University of Trieste, 34125 Trieste, Italy; 3Neurobiology Sector, International School for Advanced Studies (SISSA), 34136 Trieste, Italy

**Keywords:** periodontology, surface chemistry, bone implant interactions, cell–implant interactions

## Abstract

Background: Although the influence of titanium implants’ micro-surface properties on titanium discs has been extensively investigated, the research has not taken into consideration their whole-body effect, which may be considered possible using a combinatorial approach. Methods: Five titanium dental implants with a similar moderate roughness and different surface textures were thoroughly characterized. The cell adhesion and proliferation were assessed after adipose-tissue-derived stem cells (ADSCs) were seeded on whole-body implants. The implants’ inductive properties were assessed by evaluating the osteoblastic gene expression. Results: The surface micro-topography was analyzed, showing that hydroxyapatite (HA)-blasted and bland acid etching implants had the highest roughness and a lower number of surface particles. Cell adhesion was observed after 24 h on all the implants, with the highest score registered for the HA-blasted and bland acid etching implants. Cell proliferation was observed only on the laser-treated and double-acid-etched surfaces. The ADSCs expressed collagen type I, osteonectin, and alkaline phosphatase on all the implant surfaces, with high levels on the HA-treated surfaces, which also triggered osteocalcin expression on day seven. Conclusions: The findings of this study show that the morphology and treatment of whole titanium dental implants, primarily HA-treated and bland acid etching implants, impact the adherence and activity of ADSCs in osteogenic differentiation in the absence of specific osteo-inductive signals.

## 1. Introduction

It is estimated that in 2019, the global dental implant market was worth around 4.6 billion USD, and it is predicted to increase at an annual progression rate of 9.0% from 2020 to 2027 [1]. Healthy people typically recover quickly from dental implant surgery through the repair and regeneration of the surrounding tissues. However, people suffering from pathological conditions, such as osteoporosis or diabetes, experience delays [2].

The effectiveness of the rehabilitation process after implant placement is dependent on appropriate epithelial and bone growth, which allows the device to osseointegrate in the implantation site. The term “osseointegration” refers to the “formation of a direct contact between the dental implant and the living bone” [3]. Fast and tight osseointegration and, therefore, excellent long-term stability are mandatory for dental implants. The most important factor responsible for the stability and duration of an oral implant is likely the macro- and micro-topography of the implant. In fact, properties such as the shape, elasticity, roughness, chemical composition, electric charge, oxide type, and thickness have been demonstrated to play significant roles [4,5,6,7,8,9,10,11]. Theoretically, the surface geometry, charge and their chemical-physical modifications have to fulfill four main tasks: (1) prevent the unspecific adsorption of denatured proteins at the interface between the oral tissues and implants [12]; (2) attract differentiated or undifferentiated progenitor cells from the native tissue [13]; (3) induce native tissue or progenitor cell regeneration and differentiation [13]; and (4) guarantee an optimal load transfer to the bone [6,14]. Implants are mainly classified into four types based on their surface roughness, as determined by the arithmetical mean of the roughness area (Sa), defined as rough (Sa > 2.0 µm), moderately rough (Sa between 1.0–2.0 µm), minimally rough (Sa between 0.5–1.0 µm), or smooth (Sa < 0.5 µm) [15,16]. Many studies employing stem cells in vivo and in vitro in order to understand the mechanism of osseointegration [17,18,19] have found that implant titanium discs [20,21] with moderately rough surfaces enable better cell and bone connections than smoother or rougher surfaces [15,16,22,23,24,25,26,27,28,29]. Surface roughness is also linked to favorable effects on load transmission via the distribution of well-tolerated micro-strains, 0.25–0.50 µε [6,30,31], which also favor osteoblastic and progenitor cell differentiation [6,31,32].

Surprisingly, despite being a more realistic and clinically suitable approach [2], little is known about the impact of whole-body implants on stem cells, including their attraction or adhesion and, secondly, their proliferative and differentiation potential. Interestingly, stem cell have recently been identified as a combinatorial tissue engineering strategy that can be used to improve titanium implant osseointegration in diabetic and osteoporotic animal models [2,33].

Among stem cells, adipose-tissue-derived stem cells (ADSCs) are easily accessible and expandable in vitro, and they are capable of anti-inflammatory activity and differentiating between various lineages of paramount importance for implant dentistry, such as osteoblastic, ameloblastic, and odontoblastic lineages [34,35,36,37].

In detail, in the present research, we evaluate whether the micro-geometry and surface characteristics of five different whole-body commercially available titanium implants with a roughness between 1 and 2 μm affect the behavior of a population of ADSCs from the point of view of their adhesion, proliferation, and differentiation in vitro.

## 2. Materials and Methods

### 2.1. Cell Culture Products and Reagents

According to the ethical committee of the University of Udine, lipoaspirates were collected from healthy donors over the age of 18 after receiving informed consent and adhering to all legal criteria for confidentiality and the management of biological material.

Lipoaspirates (15 mL) were taken from the donors utilizing the tumescent technique based on an infusion of Klein’s solution followed by the recovery of the adipose tissue [34]. After that, the adipose tissue was centrifuged at 1800× *g* for 15 min to eliminate the red blood cells. The ADSCs were placed in Jocklik modified alpha-MEM medium (Gibco, Carlsbad, CA, USA) supplemented with 400 U/mL collagenase type 2 (Wortington, Columbus, OH, USA) equal to two times the volume of lipoaspirates, and the solution was incubated for 15 min at 37 °C, with gentle shaking every 2–3 min [34]. The solution was then centrifuged at 1800× *g* for 10 min [34]. The supernatant was discarded, and the pellet was resuspended and filtered through a filter to select the cell fractions smaller than 70 μm [34]. The ADSCs were plated onto 100 mm dishes (2 × 10^6^ cells per dish) and cultured in proliferation medium, adapted from Ferro F. et al., composed of α-MEM with L-glutamine 2 mmol/L, 10% fetal bovine serum (FBS), insulin 10 mg/mL, dexamethasone 10^−9^ mol/L, ascorbic acid 100 mM, EGF 10 ng/mL, and gentamicin 50 ng/mL (all from Sigma-Aldrich, Saint Louis, MO, USA) [34]. Colonies developed in the primary culture and reached near confluency within approximately one week. The ADSCs were maintained as semiconfluent to prevent cell differentiation, and approximately 80% of the medium was replaced every three days.

The stem cells derived from adipose tissue were maintained in culture medium at 37 °C, 5% CO_2_, in a humidified atmosphere. To detach the ADSCs, a bland detaching solution, namely CTC (collagenase 20 U/mL, trypsin 0.75 mg/mL, 2% heat-inactivated dialyzed chicken serum (Gibco) in Ca and Mg Hank’s Balanced Salt Solution (HBSS)), was used. The cells were maintained in culture from passages one to six (P1 to P6) on 150 mm culture plates and then were used in the experiments [34].

### 2.2. ADSC Characterization

The following antibodies were used to characterize the P6 ADSC membrane-bound cluster of differentiation stem markers: CD34, CD45, CD73, CD90, CD105, CD106 (BD, Franklin Lakes, NJ, USA), human leukocyte antigen–DR isotype (HLA-DR), IgG1 BD (BD), IgG2b, and IgG1κ (Invitrogen, Waltham, MA, USA). Briefly, the cells (2.2 × 10^6^ cells) were detached, re-suspended in incubation buffer, and incubated for 1 h at 4 °C with the antibodies. The cells were washed with PBS, centrifuged, and re-suspended in 100 μL wash buffer before being analyzed. A threshold was established for the forward and side scatter dot plots to exclude the cellular debris, and 20,000 events were analyzed using the Becton-Dickinson (BD) FACS Canto flow cytometer and FlowJo (BD) software.

### 2.3. Multi-Differentiation Assays

The ADSCs at P6 were differentiated through the osteoblastic, adipocyte, and chondroblastic lineages using the following media composition: F-12 Coon’s modified/Ambesi’s modified medium (Gibco) supplemented with dexamethasone 100 nM, b-glycerophosphate 1 mM, Ca^2+^ 1.2 mM, Mg^2+^ 0.6 mM, glucose 2 g/L, vitamin K2 1 mM, vitamin D3 5 nM, gentamicin 50 ng/mL, FBS 0.5% (all from Sigma-Aldrich, Saint Louis, MO, USA), retinoic acid 250 nM, 17-β-estrogen 1 mM, and calcitonin 1 nM (all from MP-Biomedicals, Santa Ana, CA, USA). The following components were used to produce the adipogenic induction and maintenance media: DMEM high glucose (HG) (Gibco), 3-isobutyl-1-methyl-xanthine 25 µM, insulin 10 µg/mL, dexamethasone 1 µM, indomethacin 200 µM, gentamicin 50 ng/mL, FBS 10%, (Sigma-Aldrich), DMEM (HG), FBS 10%, insulin 10 µg/mL, and gentamicin 50 ng/mL. The chondrogenic medium composition was as follows: DMEM (HG), dexamethasone 100 nM, ascorbic acid 50 µg/mL, L-proline 40 µg/mL, insulin 5 µg/mL, transferrin 5 µg/mL, selenous acid 5 ng/mL, linoleic acid 5.35 µg/mL, transforming growth factor-3 (TGFβ-3) 10 ng/mL, sodium pyruvate 1 mM, and gentamicin 50 ng/mL (all from Sigma-Aldrich). At day 20, Alizarin Red S (Sigma-Aldrich) staining was used to validate the osteogenic capacity of the ADSCs. Oil Red O (Sigma-Aldrich) was used to measure the adipogenic differentiation on day 21. After 21 days in culture, the degree of chondrogenesis differentiation was assessed by safranin O staining (Sigma-Aldrich).

### 2.4. Characteristics of the Dental Implants

Five different titanium dental implants were compared in this study, as follows: plasma spray; laser; HA-blasted with hydroxyapatite (HA) and bland acid etching; double acid etching; and HA-blasted with HA followed by “new” patented treatment (Table 1). The plasma-spray-treated implant is made of grade 4 titanium with the overall nominal chemical composition shown in Table 1 [38], and the device is machined automatically before being packaged. The laser-treated implant is made of grade 4 titanium, with the overall nominal chemical composition shown in Table 1 [38]. The implant surface is created through the use of a DPSS Nd Qswitching laser source (solid-state, diode-pumped laser, DPSS, with an Nd source in the Q-sw regime). With this technique, the material is removed from the surface in the form of vapor, and this “cold” ablation has a high repeatability. The DPSS Nd Q-sw laser technology allows us to create perfectly reproducible micrometric porosities in terms of the shape, diameter, and depth, as well as the distribution and spacing. The HA-blasted implant with HA and bland acid etching is made of Ti6Al4V alloy, with the overall nominal chemical composition shown in Table 1 [38]. The implant surface is obtained by means of a blasting process using HA, followed by treatment with a weak acid to remove the excess micro-granules. The use of HA micro-granules during the blasting process aims to obtain a rougher surface. The double-acid-etching-treated implant is made of grade 4 titanium, with the overall nominal chemical composition shown in Table 1 [38]. The implant surface is obtained via a thermal etching process with hydrochloric acid and sulfuric acid. This process gives rise to numerous irregularities on the surface at distances of about 1–3 μm, distributed between the peaks. Moreover, the presence of micro-pores of a size of about 1–2 μm increases its surface complexity. The HA-blasted implant subsequently subjected to a “new” patented treatment is made of Ti6Al4V alloy, with the overall nominal chemical composition shown in Table 1 [38]. The description of the surface treatment is protected by a patent.

### 2.5. Dental Implant and Surface Characterization

Morphological analyses were executed using a scanning electron microscope Quanta250 (SEM, FEI, Hillsboro, OR, USA) in a high vacuum and in secondary electron mode, with 30 kV of tension. The operational distance was fixed in order to acquire an appropriate magnification. The samples were secured on aluminum stubs covered with carbon double-sided tape and consequently gold-sputtered with a Sputter Coater K550X (Emitech, Quorum Technologies Ltd., Laoughton, UK).

SEM photomicrographs (*n* = 3, magnification 1500×) were binarized and thresholded before being processed using ImageJ and the nearest distances (ND) plugin (http://imagej.nih.gov, USA, accessed on 2 September 2022) to determine the average size of, and distance between, the implant surface particles. Differences in roughness between the implants were assessed using SEM photomicrographs and the SurfCharJ plugin [13] to evaluate the following parameters: the Rq (root mean square deviation), Rsk (skewness of the assessed profile), Rku (kurtosis of the assessed profile), and Ra (arithmetical mean deviation). Imagej was also used to obtain the surface plot and 3D profile differences of the implants [39].

### 2.6. Dental Implant Seeding with ADSCs

Each dental implant (*n* = 3) was placed in the vertical position on a 96 multi-well plate (Gibco) and placed onto a 150 mm plate to maintain its sterility. The wells were filled with a known amount of proliferation medium (200 μL) and 1.25 × 10^6^ ADSCs. The medium was changed on day one to remove non-adherent cells and also on day three (Figure 1).

### 2.7. Cell Adhesion and Proliferation

The cell adhesion was assessed on day one. On days three and seven, the cell proliferation on the implant surface was measured (*n* = 3). The cell detachment from the implants was performed after transferring the implants to 1.5 mL conical tubes containing CTC for 10 min at 37 °C in a humid atmosphere with 5% CO_2_ (Figure 1). The Neubauer chamber was used to count the cells in triplicate at each time point.

### 2.8. Gene Expression Analysis

The expression of the osteoblastic markers was quantified on days one, three, and seven in triplicate by PCR using the following primers: alkaline phosphatase (ALP) GCAGGCAGGCAGCTTCAC TCAGAACAGGACGCTCAGG, 496 bp, 60.5 °C, NM_000478.3; osteocalcin (Osc) TCACACTCCTCGCCCTATTG CTAGACCGGGCCGTAGAAG, 293 bp, 58 °C, NM_199173.3; osteonectin (Osn) CACAAGCTCCACCTGGACTA GAATCCGGTACTGTGGAAGG, 525 bp, 58 °C, NM_003118.2, and collagen Type I (Coll-I) TAAAGGGTCACCGTGGCT CGAACCACATTGGCATCA, 355 bp, 60 °C, NM_000088.3. RNA polymerase type II GCACCACGTACACCAATG GTGGGGCTGCTTTAACCA, 350 bp, 56 °C, NM_000937, was used as housekeeping gene (MWG Eurofins, Ebersberg, Germany).

### 2.9. Statistical Analysis

The morphologic and structural variables of the implants, cell counts, and gene expression are expressed as the mean ± standard deviation. Statistical significance was evaluated by analysis of variance (ANOVA) followed by Fisher’s or Bonferroni’s post hoc test. The paired *t*-test was used to compare the data obtained from the related groups. SPSS 23.0 (IBM, Armonk, NJ, USA) was used to analyze the data, and statistical significance was defined as *p* ≤ 0.05 and *p* ≤ 0.001.

## 3. Results

### 3.1. ADSC Isolation, Characterization, and Stemness Potential Assessment

We detailed the ADSC isolation methodology in Figure 1, along with the photomicrographs of the ADSCs at P3 following their isolation from the lipoaspirates obtained from healthy donors using the tumescent technique (Figure 1).

The fingerprints of the common stemness-related membrane proteins were validated using flow cytometry. The results are expressed as the mean ± SD of the three independent experiments, with * indicating significance at *p* ≤ 0.05 (ANOVA followed by Bonferroni’s post hoc test), and they confirmed that the stem cell markers CD73 (98.9% ± 1.03), CD90 (99.9% ± 0.09), and CD105 (99.8% ± 0.15) were highly present on the ADSCs’ membranes, whereas the hematopoietic CD34 (0.82% ± 0.34), and CD45 (0.087% ± 0.03), and finally CD106 (0.71% ± 0.63) were not (Figure 2A).

The ADSCs were then differentiated through the adipocyte, osteoblastic, and chondroblastic lineages to confirm their multi-lineage potential as a result of three independent experiments. In comparison with the control undifferentiated cells, the differentiated ADSCs developed through the adipocyte lineage, as evidenced by the presence of lipid droplets revealed by Oil Red O staining (red, Figure 2B). Following their osteoblastic differentiation, the ADSCs were stained with alizarin red, displaying a high level of calcium and, hence, osteoblastic differentiation when compared to the undifferentiated cells (red, Figure 2C). Finally, safranin O staining revealed chondroblastic differentiation in the ADSCs when compared to control cells, indicating the existence of a central core of glycosaminoglycans (violet, GAGs) following three-dimensional chondroblastic differentiation (Figure 2D).

### 3.2. Implant Morphological Characterization

The SEM micro-photographs revealed a heterogeneous morphology of the surfaces of the implants (Figure 3). The textures of the HA-blasted and bland acid etching implants, as well as the HA-blasted implants with the “new” under-patent treatment, were quite similar, revealing that many HA particles from the blasting treatment adhered to the surface, having a shattered appearance and emerging from the surface. The plasma spray and double-acid-etched surfaces were less irregular than the previous HA-treated surfaces. The surface texture of the laser-treated implant was the most uniform, with occasional voids resembling Howship’s lacunae. (Figure 3). The surface area of the implants, expressed in mm^2^ as the mean ± SD of the three independent experiments, with * indicating significance at *p* ≤ 0.05 (ANOVA followed by Bonferroni’s post hoc test), was as follows: plasma spray (0.097 mm² ± 0.004), laser (0.108 mm² ± 0.01), HA-blasted and bland acid etching (0.12 mm² ± 0.032), double acid etching (0.12 mm² ± 0.015), and HA-blasted “new” (0.081 mm² ± 0.023) (*p* ≤ 0.05).

The micro-architecture of the implants, including the size, distribution, and density of the irregularities, as well as the roughness, is widely acknowledged as having a substantial influence on their mechanical characteristics and interaction with cells [40]. As a result, we analyzed the mean surface particle area, distance, and the spacing between each particle to provide a quantitative measurement of how uniformly the features of interest were distributed across the surface [40] (Figure 4A–E, Table 2).

The results, expressed as the mean ± SD, of the three independent experiments, with * indicating significance at *p* ≤ 0.05 (ANOVA followed by Bonferroni’s post hoc test), showed that the HA-blasted and bland acid etching implants (222 ± 29, *p* ≤ 0.05) had the lowest number of surface particles and, despite having the largest mean area of the surface particles (20.69 μm ± 57.53), did not differ significantly from all the other implants (Figure 4F). The HA-blasted and bland acid etching implants also showed the highest mean distance between particles (20.69 μm ± 57.53, *p* ≤ 0.05), the nearest particle distance (4.49 μm ± 2.18, *p* ≤ 0.05), and the greatest average wall thickness between particles (3.13 μm ± 2.54, *p* ≤ 0.05) (Figure 4G–I).

In addition, according to ISO 4287/2000, we analyzed some well-known surface roughness parameters [13]. Differences in roughness, expressed as the mean ± SD of three independent experiments, with * indicating significance at *p* ≤ 0.05 (ANOVA followed by Bonferroni’s post hoc test), were found between the implants regarding all the parameters, including the arithmetical mean deviation (Ra), root mean square deviation (Rq), skewness of the assessed profile (Rsk), and kurtosis of the assessed profile (Rku).

In the HA-blasted and bland acid etching implants, the parameters of the Ra (1.16 μm ± 0.04) and Rq (1.72 μm ± 0.03, *p* ≤ 0.05) (Figure 5H) were the highest with regard to all the other implants (*p* ≤ 0.05) (Table 3 and Figure 5G, *p* ≤ 0.001). In contrast, the HA-blasted and bland acid etching implant had the lowest values for the roughness-associated parameters, the Rsk and Rku, respectively (1.48 μm ± 0.02 and 2.19 μm ± 0.07), differing significantly from all the other implants (*p* ≤ 0.05) (Table 3 and Figure 5I,J).

To overcome any issues related to the seeding approach, we report the ratio of the adherent cells retrieved at the three time points to the total number of seeded cells at day zero (1.25 × 106 cells) and the surface area of the implants (Figure 6). The results, expressed as the mean ± SD of three independent experiments, with * indicating significance at *p* ≤ 0.001 (ANOVA followed by Bonferroni’s post hoc test and the paired *t*-test), highlighted the following differences.

Indeed, on day one, the HA-blasted and bland acid etching (366 ± 77 cells/mm^2^) implants had the highest adhesion, followed by the plasma spray (315 ± 70 cells/mm^2^), HA-blasted “new” (266 ± 22 cells/mm^2^), double acid etching (212 ± 40 cells/mm^2^), and, lastly, laser-treated (122 ± 32 cells/mm^2^) implants (Figure 6A–E) (*p* ≤ 0.001).

By day three, we discovered an increase in the cell number only on the laser-treated (185 ± 55 cells/mm^2^) implants, whereas the number fell in the case of all the other implants, including the HA-blasted “new” (243 ± 91 cells/mm^2^), HA-blasted and bland acid etching (155 ± 42 cells/mm^2^), plasma spray (118 ± 38 cells/mm^2^), and double acid etching implants (56 ± 17 cells/mm^2^) (Figure 6A–E) (*p* ≤ 0.05).

We discovered a proliferative phase in the double acid etching (83 ± 25 cells/mm^2^) implants, whereas the laser-treated implants (193 ± 57 cells/mm^2^) exhibited a constant number of cells, and the HA-blasted “new” (197 ± 74 cells/mm^2^), HA-blasted and bland acid etching (139 ± 38 cells/mm^2^) and plasma-spray (83 ± 26 cells/mm^2^)-treated implants showed a significant reduction in the adhering cell number by day seven (Figure 6A–E) (*p* ≤ 0.05).

When the adherent cells were evaluated at the same time point across the different implant surfaces, and the results were expressed as the mean ± SD of the three independent experiments, with §, *, ^, $ indicating significance at *p* ≤ 0.001 (ANOVA followed by Bonferroni’s post hoc test and the paired *t*-test), it was obvious that the plasma-spray-treated surfaces, after 24 h, varied substantially from the laser-treated surface alone (*p* < 0.001), whereas the laser surface was distinct from all the other treated implanter surfaces (*p* < 0.001). The HA-blasted and bland acid etching, double acid etching, and HA-blasted “new” treatments differed considerably on all the surfaces except one, which was the plasma spray surface (Figure 6F and Table 4).

At the three-day time point, the plasma spray treatment differed with respect to all but one implant treatment, which was the double acid etching treatment. At the same time point, the laser, HA-blasted and bland acid etching, and the HA-blasted “new” surfaces differed significantly from all the surfaces tested (respectively *p* = 0.010, *p* = 0.003, *p* < 0.001). The double-acid-etched surface differed only from the blasted surfaces (*p* < 0.001) (Figure 6F and Table 4).

Later, on the seventh day, the plasma spray treatment varied considerably from the laser treatment and the HA-blasted “new” treatment (*p* < 0.001) (Figure 6F and Table 4). The laser treatment differed from the plasma spray treatment (*p* ≤ 0.001) and the double acid etching treatment (*p* = 0.049). On day seven, the HA-blasted and bland acid etching surface did not vary from any of the treated implants. The HA-blasted “new” treatment diverged from the plasma spray (*p* < 0.001) and double acid etching (*p* = 0.001) treatments, whereas the double acid etching differed solely from the laser (*p* = 0.049) and the HA-blasted “new” treatments (*p* = 0.001) (Figure 6F and Table 4).

### 3.3. Osteoblastic Induction

To validate the effectiveness of the experimental seeding strategy, we evaluated the expression of various well-known extracellular matrix (ECM) osteoblastic markers, normalizing their values, expressed as the mean ± SD of three independent experiments, with * indicating significance at *p* ≤ 0.05 (ANOVA followed by Bonferroni’s post hoc test), for the adherent cell number and surface area. By increasing the expression of Osn, Coll-I, and ALP with respect to the housekeeping gene in the control cells, all of the treatments and surfaces were able to drive osteoblastic differentiation. The HA-blasted “new” and laser-treated surfaces elicited the greatest ALP and Coll-I expressions when compared to the other implant types at the 24 h time point (Figure 7A,B; Table 5).

By comparison, the greatest Osn expression was achieved under the effects induced by the double-acid-etched surface, while the HA-blasted “new” and laser treatments demonstrated a lower ability to induce Osn (Figure 7D; Table 5). Additionally, Osc was not present in all the samples at 24 h, nor was it present at three days (data not shown).

After three days, the HA-blasted “new”, laser-, and plasma-spray-treated surfaces elicited the greatest ALP, Coll-I, and OSN expressions when compared to the other implant types at the 24 h time point (Figure 7A–C; Table 5). On the contrary, the HA-blasted and bland acid etching and the double acid etching surfaces elicited the smallest production of Osn, ALP, and Coll-I messengers (Figure 7A–C; Table 5).

By day seven, the HA-blasted-“new”-treated surface still elicited the highest levels of Coll-I and OSN, but the laser and plasma spray surfaces exhibited less inductive activity for these extracellular markers (Figure 7A,C; Table 5). The ALP induction reached a similar peak by day seven in the laser and double acid etching implants, while the HA-blasted “new” implant showed a reduced effect on the ALP production (Figure 7A; Table 5). Interestingly, at day seven, even though the HA-blasted and bland acid etching surface was not as inductive with respect to all the other surfaces, it was the only surface able to induce the late osteoblastic differentiation marker Osc (Figure 7D; Table 5).

## 4. Discussion

In simple terms, the effectiveness of dental implants depends on the creation of a barrier that is capable of both sealing the underlying osseous structures and integrating the body of the implant. It is well-known that the tissues interacting with the implant surface are essentially three in number. The first, closely bound to the implant, is poorly cellularized, and its ECM is composed of large and dense bundles of thick Coll-I fibers that contribute to the mechanical resistance and stability of the implants [41,42]. The second is relatively rich in fibroblasts, with a large number of secretory components, and is structurally formed of Coll-I fibers that are heavily associated with collagen type III [41,42]. The third is the bone, which is mostly formed of inorganic mineral material HA with interspersed extracellular proteins, such as Coll-I, Osc, Osn, osteopontin, ALP, and a few scattered osteocytes [41,42].

In this study, we used whole titanium dental implants with moderately rough surfaces, instead of titanium disks [17,20,23,43,44], and ADSCs, chosen because of their ease of separation, as well as their stemness properties [34], to deliver a more realistic approach and to assess the feasibility of a hypothetical combinatorial strategy.

The results revealed that, whereas the topological properties of the implant surfaces under consideration were similar [17,19,25,45], there were substantial differences in the roughness, with the HA-blasted and bland acid etching implants exhibiting the highest value. Furthermore, while the HA-blasted and bland acid etching implants had the fewest surface particles, they also had the lowest density with respect to the other implants.

The stress caused by the mastication loads on a dental implant creates dynamic strains on the surrounding tissues, thus affecting the rehabilitation process [6,14,31]. In view of this, the surface particles and roughness guarantee a differential load transfer depending on their physical and chemical characteristics [6,14,31]. Therefore, the presence of a configuration with the highest roughness and reduced number of particles on the HA-blasted and bland acid etching implants may have a considerably better impact on the adhesion and load transfer during rehabilitation [19,23].

A large body of evidence asserts that many contaminants, being either metallic or nonmetallic, such as C, Mg, Fe, Al, Ca, P, Sr, and F, are introduced onto the implant surface voluntarily (commercially pure titanium grades and titanium alloys) or, in spite of the strict control measures, during the manufacturing process or handling [19,46,47]. We previously demonstrated that machined and laser micro-patterned treatments showed no traceable surface impurity or modification, whereas “sandblasting” introduced elemental traces of C, Fe, Al, and O [48], thus chemically impacting the surface, changing its composition, and influencing the tissue and cell activity [19,46,47]. Another factor that has to be taken into account is the fact that any approach to modifying the surface roughness alters the surface chemistry, finally altering the protein adsorption/adhesion, i.e., Ca^2+^ and Mg^2+^ [12].

As a result, it is possible to hypothesize that the procedures used for HA-blasted and bland acid etching implants modify both the chemical [19,46,47] and topographical properties, such as the roughness or particle number [19,23], ultimately favoring cell adhesion [12].

Following the initial adhesion, which reached a peak for the HA-blasted and bland acid etching implants, a non-significant proliferative phase occurred for all but one implant, namely, the laser treated implants on day three and the double acid etching implants on day seven. This stands in contrast to previous studies on titanium discs [7,18,20], and it might be due to the low cell density or the whole-body surface via the presence of solubilized metallic ions with negative effects, which cause inflammation and cytotoxicity as their concentration rises [49,50].

On the other hand, the findings suggest that the surface treatment and topography of the whole-body implants (roughness [18,26,27], the surface particle density and characteristics [32], and, potentially, the chemical composition [19,47,50]) have active impacts on ADSC osteoblastic marker synthesis. In support of this view, many studies employing stem cells in vivo and in vitro have found that dental implants [17,18,19,20,21] with moderately rough surfaces allow for better osseointegration [15,16,17,18,19,20,21,22,23,24,25,26,27,28,29] and reduced marginal bone loss [24,51] compared to smoother or rougher surfaces.

It is known that once cells adhere to biomedical materials, their interaction elicits profound responses within the cells that, depending on how they are perceived, might result in proliferation, differentiation, and, therefore, survival, or apoptosis and, thus, cell death [23,26,27,52,53,54,55].

The contact/interaction between the actin cytoskeleton and focal adhesion proteins and the implant surface has been identified as the critical regulator of the integration process [52]. Indeed, the integrin-mediated interaction between the extracellular matrix proteins, such as fibronectin, vitronectin, osteonectin, and collagen-I, and the implant surface has been documented to regulate cell adhesion, differentiation, and survival via programmed cell death (apoptosis) [23,26,27,52,53,54,55]. Accordingly, we assume that, following the adhesion, the cytoskeleton of the ADSCs underwent considerable reorganization, resulting in the intracellular signal transmission to the cytoplasm and nucleus [23,52]. The signals, in turn, triggered a variable degree of differentiation, coincident with the expression of increased levels of ALP, Osn, and Coll-I, starting on day one and lasting until day seven. Even though neither the HA-blasted and bland acid etching nor the HA-blasted “new” implants produced significantly more ALP, Osn, or Coll-I than the other treatments, only the HA-blasted and bland acid etching implants could induce the expression of the late osteoblastic marker (Osc). In general, our findings suggest that the surface topography (similar roughness or particle number and distribution) of the HA-treated surfaces promotes the ADSCs’ differentiation and, possibly more, durable osseointegration due to the better load distribution through the surrounding bone during rehabilitation [19,23]. It is also plausible to speculate that the processes utilized for the HA-blasted implants changed the chemical composition of the surface, i.e., by adding Ca^2+^, P^5+^, Na^+^, S^6+^, Si^4+^, and O^2−^ ions [19,46,47], thus promoting the differentiation of the ADSCs.

## 5. Conclusions

In summary, we discovered that the surface roughness and treatment composition of dental implants play important roles in the adhesion and differentiation of ADSCs grown on whole-body titanium implants but not in their proliferation.

Significant progress has been achieved in the application of material engineering to the study and modulation of a variety of restorative and pathological disorders [2,56,57,58,59,60,61,62], and preclinical attempts have been made to combine material engineering with stem cells [2]. Based on our findings, we suggest that HA-blasted titanium implants combined with adherent ADSCs, which are preconditioned [57,63] to increase their in vivo potential, might be helpful in clinical settings for people with systemic conditions, such as diabetes and osteoporosis [2]. In such a way, it may be feasible to promote quicker osseointegration and increase the patient’s rapid tissue regeneration and bone production, therefore improving their rehabilitation.

However, different seeding concentrations, cytotoxicity tests, longer culture intervals, and pre-clinical studies are still required in order to fully grasp the advantages of HA-blasted implants over the other types and, therefore, to determine the most efficient and suitable surface to employ in conjunction with stem cells.

## Figures and Tables

**Figure 1 jfb-13-00206-f001:**
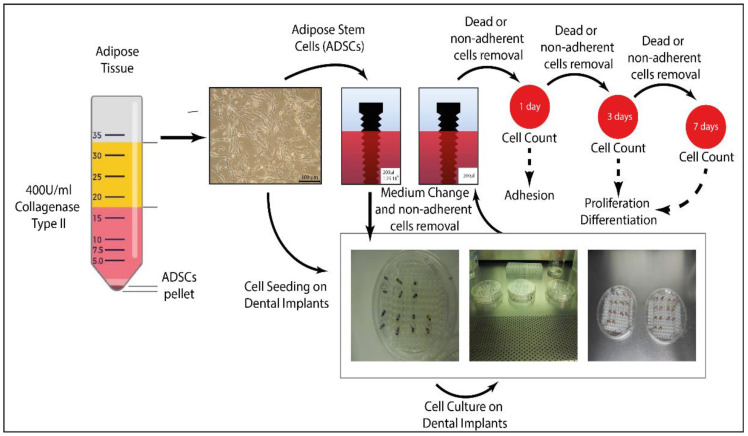
Schematic representation of the seeding and culture phases of the cells on the commercial implants. ADSCs were isolated from lipoaspirates and cultured. Cells were then diluted in proliferation medium and seeded onto whole-body dental implants. Adhesion was assessed on day one, and the proliferation and osteoblastic differentiation were assessed on days three and seven.

**Figure 2 jfb-13-00206-f002:**
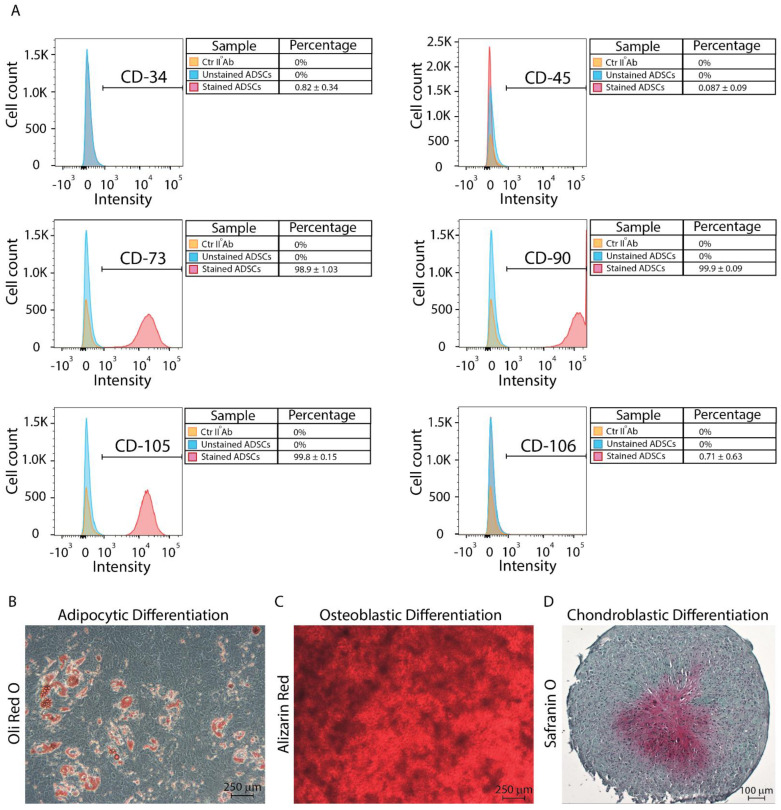
Stemness and differentiation potential assessment. (**A**) The ADSC stemness was confirmed by verifying the expression of CD73, CD90, and CD105 and the reduced presence of CD34, CD45, and CD106. Differentiation was assessed by inducing the ADSCs through the adipocytic (**B**), osteoblastic (**C**), and chondroblastic (**D**) lineages. Scale bars: 250−100 μm. Results are expressed as mean ± SD of three independent experiments. Abbreviations: cluster of differentiation (CD).

**Figure 3 jfb-13-00206-f003:**
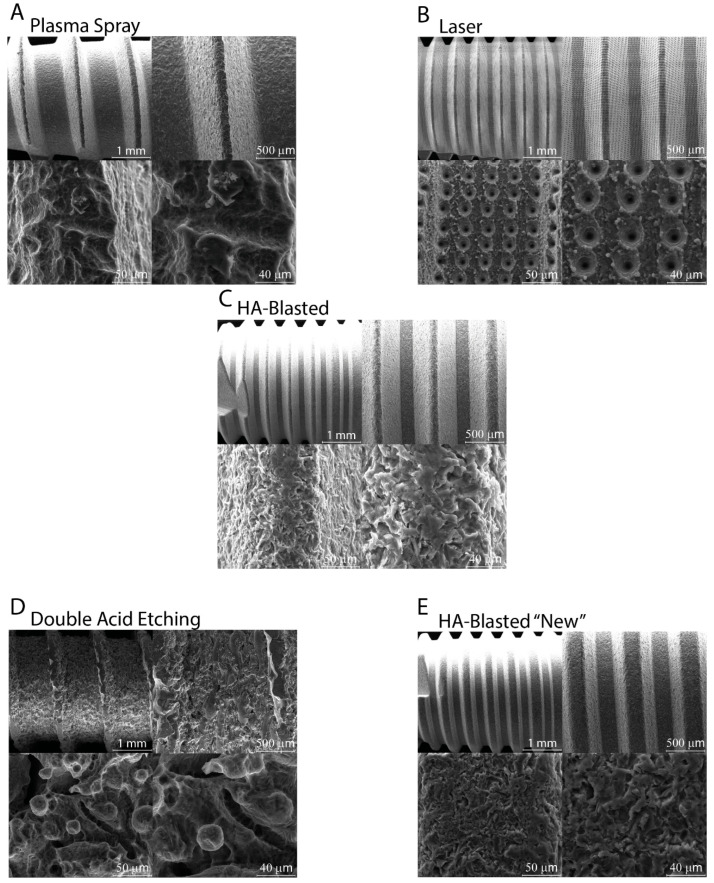
Micro-surface characteristics are presented in the SEM photomicrographs. The surface textures of the HA-blasted and bland-acid-etching implants (**C**), as well as the HA-blasted implants with the “new” under-patent treatment (**E**), were highly comparable, suggesting that many HA particles from the blasting process fixed to the surface and became part of it. The plasma spray (**A**) and double-acid-treated (**D**) surfaces were less irregular than the previous one. The laser-treated (**B**) implant exhibited the most uniform surface texture, with regular voids that resembled Howship’s lacunae. Scale bars: 1 mm, 500-50-40 μm.

**Figure 4 jfb-13-00206-f004:**
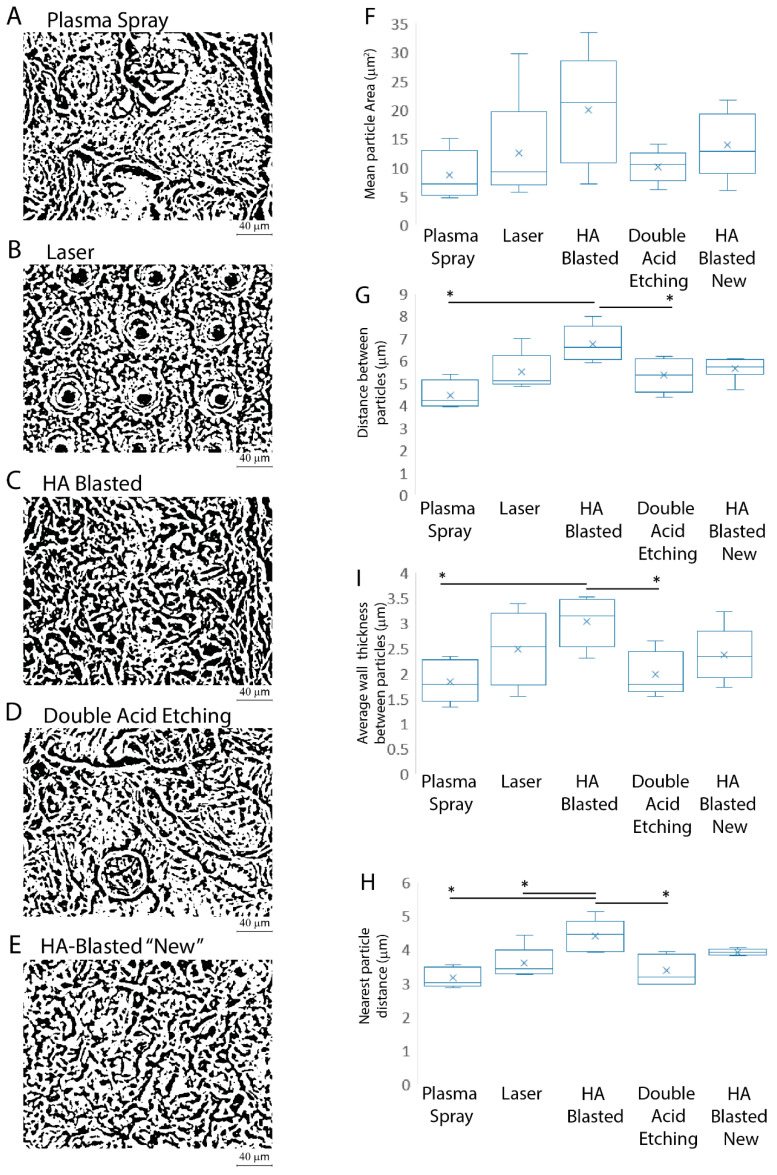
Summary of the surface particle characteristics. The plasma spray (**A**), laser (**B**), HA-blasted and bland acid etching (**C**), double acid etching (**D**), and HA-blasted “new” (**E**) implants were analyzed to provide a quantitative measurement of how uniformly the mean surface particle area (**F**), distance (**G**), and the spacing between each particle (**I**) and its closest neighbors’ (**H**) features of interest were distributed across the surface. Results are expressed as mean ± SD of three independent experiments, with * indicating significance at *p* ≤ 0.05 (ANOVA followed by Bonferroni’s post hoc test).

**Figure 5 jfb-13-00206-f005:**
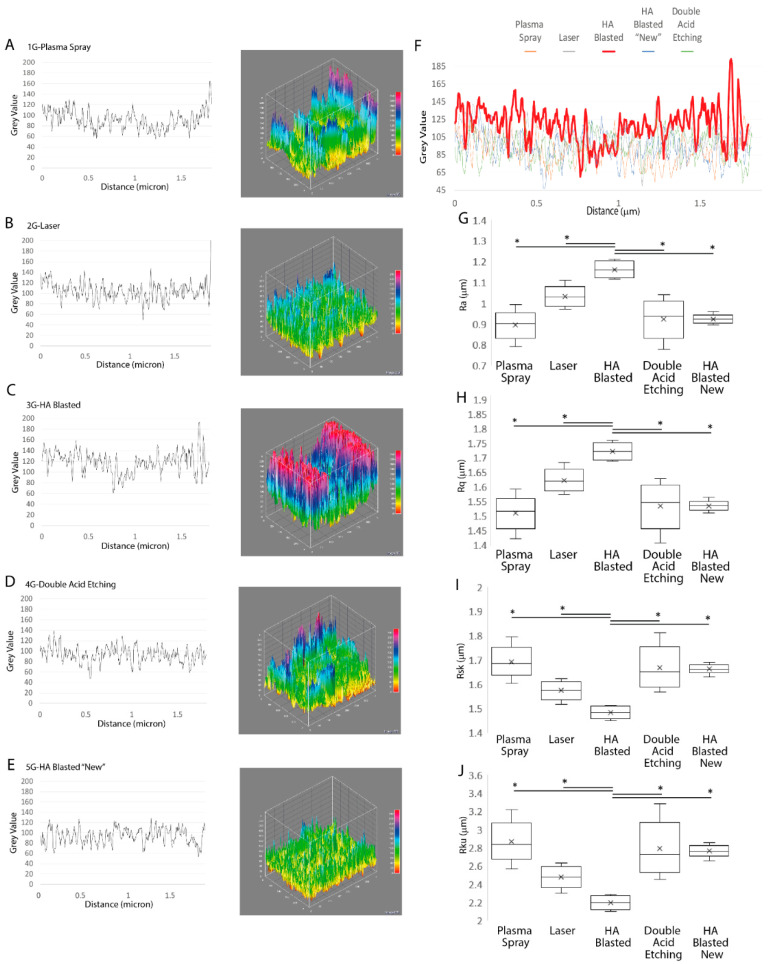
Interfacial roughness profile graphs and 3D profiles. The corresponding roughness profiles of the implants and the specific values used to generate the 3D surface roughness graphs, including the plasma spray (**A**), laser (**B**), HA-blasted and bland acid etching (**C**), double acid etching (**D**), and HA-blasted “new” (**E**) implants, were calculated from the height and distance of the peak profiles in order to highlight the differences in roughness (**F**) found between the implants’ surfaces tested in this study, including the Ra (**G**), Ru (**H**), Rsk (**I**), and Rku (**J**). Results are expressed as mean ± SD of three independent experiments, with * indicating significance at *p* ≤ 0.05 (ANOVA followed by Fisher’s post hoc test). Abbreviations: arithmetical mean deviation (Ra), root mean square deviation (Rq), skewness of the assessed profile (Rsk), kurtosis of the assessed profile (Rku).

**Figure 6 jfb-13-00206-f006:**
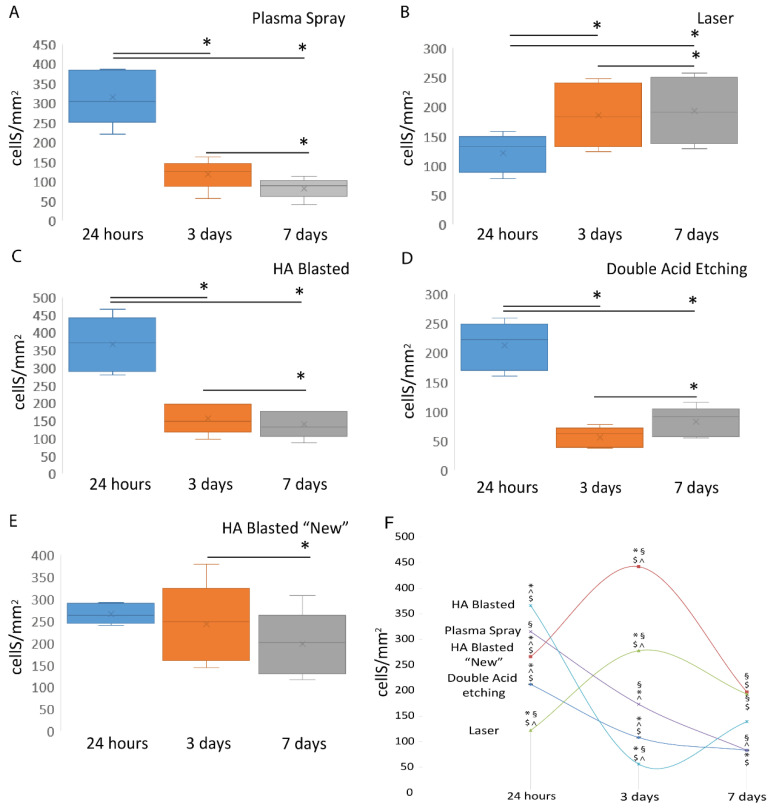
Summary of the ADSCs’ adhesion to the tested dental implants. Graphs of the cell count after 24 h and after three and seven days of the ADSCs’ culture on the whole surface of the commercial implants, including the plasma spray (**A**), laser (**B**), HA-blasted and bland acid etching (**C**), double acid etching (**D**) and HA-blasted “new” implants (**E**). Comparison of the adhesion and proliferation outcomes between the different implants at the same time point (**F**). Results are expressed as mean ± SD of three independent experiments, with §, *, ^, $ indicating significance at *p* ≤ 0.001 (ANOVA followed by Bonferroni’s post hoc test and paired *t*-test).

**Figure 7 jfb-13-00206-f007:**
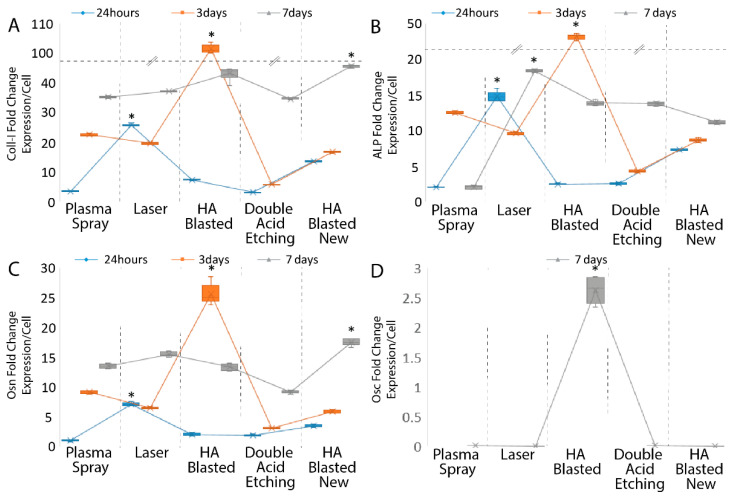
Summary of the ADSC differentiation induction following seeding onto the tested dental implants. Early and late osteoblastic markers, Coll-I (**A**), ALP (**B**), Osn (**C**), and Osc (**D**), and the expression of ADSCs cultured on the commercially available dental implants after 24 h and three and seven days (**A**–**D**). Results are expressed as mean ± SD of three independent experiments, with * indicating significance at *p* ≤ 0.001 (ANOVA followed by Bonferroni’s post hoc test). Abbreviations: alkaline phosphatase (ALP); osteocalcin (Osc); osteonectin (Osn); collagen type 1 (Coll-I).

**Table 1 jfb-13-00206-t001:** Summary of the characteristics of the tested implants. Five different titanium-formed implants with a roughness between 1 and 2 μm and different surface treatments were tested in this study. Nominal chemical composition of the titanium and titanium alloy implants.

Surface Treatment	Plasma Spray	Laser	HA-Blasted and Bland Acid Etching	Double Acid Etching	HA-Blasted “New” under Patent
Composition	Grade 4 titanium	Grade 4 titanium	Ti6Al4V	Grade 4 titanium	Ti6Al4V
Nominal chemical composition [38]	Ti 99%	Ti 99%	Ti 90%	Ti 99%	Ti 90%
	Fe 0.3%	Fe 0.3%	Fe 0.25%	Fe 0.3%	Fe 0.25%
	O 0.4%	O 0.4%	O 0.2% max	O 0.4%	O 0.2% max
	C 0.1%	C 0.1%	C 0.0%	C 0.1%	C 0.0%
	N 0.05%	N 0.05%	N 0.0%	N 0.05%	N 0.0%
	H 0.015%	H 0.015%	H 0.0%	H 0.015%	H 0.0%
	Al 0.0%	Al 0.0%	Al 6.4%	Al 0.0%	Al 6.4%
	V 0.0%	V 0.0%	V 4.12%	V 0.0%	V 4.12%

**Table 2 jfb-13-00206-t002:** Summary of the surface particle characteristics. The mean surface particle area, distance, and the spacing between each particle and its closest neighbors were analyzed to provide a quantitative measurement of how uniformly the features of interest were distributed across the surface.

	Plasma Spray	Laser	HA-Blasted	Double Acid Etching	HA-Blasted “New”
Particle area (μm^2^)	8.59 ± 23.18	11.91 ± 41.22	20.69 ± 57.53	10 ± 22.2	13.19 ± 29.09
Mean distance between particles (μm)	4.55 ± 1.54	5.65 ± 2.16	6.95 ± 2.5	5 ± 1.45	5.8 ± 1.63
Nearest Particle Distance (μm)	3.14 ± 1.27	3.57 ± 1.75	4.49 ± 2.18	3.35 ± 1.27	3.92 ± 1.61
Average Wall Thickness between particles (μm)	1.82 ± 1.37	2.52 ± 2.5	3.13 ± 2.54	1.98 ± 1.51	2.37 ± 1.82
Particles per surface	518 ± 67	316 ± 39	222 ± 29	464 ± 78	337 ± 94

**Table 3 jfb-13-00206-t003:** Roughness-related parameters analyzed using the ImageJ plugin (sample length 40 μm). Abbreviations: arithmetical mean deviation (Ra), root mean square deviation (Rq), skewness of the assessed profile (Rsk), kurtosis of the assessed profile (Rku).

	Plasma Spray	Laser	HA-Blasted	Double Acid Etching	HA-Blasted New
**Ra (μm)**	0.89 ± 0.07	1.03 ± 0.05	1.16 ± 0.04	0.92 ± 0.09	0.92 ± 0.02
**Rq (μm)**	1.5 ± 0.06	1.62 ± 0.04	1.72 ± 0.03	1.53 ± 0.08	1.53 ± 0.02
**Rsk (μm)**	1.69 ± 0.07	1.57 ± 0.04	1.48 ± 0.02	1.66 ± 0.09	1.66 ± 0.02
**Rku (μm)**	2.86 ± 0.24	2.47 ± 0.12	2.19 ± 0.07	2.78 ± 0.31	2.76 ± 0.07

**Table 4 jfb-13-00206-t004:** Comparison of different implants at various time points. An overview of the data analysis comparison of the various implants at different time points. The data are significant with *p* ≤ 0.001 (ANOVA followed by Bonferroni’s post hoc test).

Reference Implant	Implant Type	24 h	3 Days	7 Days
Plasma Spray	Laser	<0.001	0.01	<0.001
	HA-Blasted	0.152	0.003	0.125
	Double Acid Etching	0.234	0.054	0.999
	HA-Blasted “New”	0.197	<0.001	<0.001
Laser	Plasma Spray	<0.001	0.01	<0.001
	HA-Blasted	<0.001	<0.001	0.153
	Double Acid Etching	<0.001	0.034	0.049
	HA-Blasted “New”	<0.001	<0.001	999
HA-Blasted	Plasma Spray	0.152	0.003	0.125
	Laser	<0.001	<0.001	0.153
	Double Acid Etching	<0.001	<0.001	0.075
	HA-Blasted “New”	<0.001	<0.001	0.113
Double Acid Etching	Plasma Spray	0.234	0.054	0.999
	Laser	<0.001	0.034	0.049
	HA-Blasted	<0.001	<0.001	0.075
	HA Blasted “New”	<0.001	<0.001	0.001
HA-Blasted “New”	Plasma Spray	0.197	<0.001	<0.001
	Laser	<0.001	<0.001	0.999
	HA-Blasted	<0.001	<0.001	0.113
	Double Acid Etching	<0.001	<0.001	0.001

**Table 5 jfb-13-00206-t005:** Comparison of mRNA expression on different implants at various time points. An overview of the data analysis comparison of the mRNA (ALP, Osn, Coll-I and Osc) expressed by ADSCs on various implants at different time points. Abbreviations: ALP, alkaline phosphatase; Osn, osteonectin; Coll-I, collagen type I; Osc, osteocalcin.

		Coll-I			ALP			Osn		
Reference Implant	Implant Type	24 h	3 days	7 days	24 h	3 days	7 days	24 h	3 days	7 days
Plasma Spray	Laser	<0.001	0.001	<0.001	<0.001	<0.001	<0.001	<0.001	<0.001	<0.001
	HA-Blasted	<0.001	0.002	<0.001	<0.001	<0.001	<0.001	1	<0.001	<0.001
	Double Acid Etching	<0.001	<0.001	<0.001	0.008	<0.001	<0.001	<0.001	<0.001	<0.001
	HA-Blasted “New”	<0.001	0.314	<0.001	0.004	<0.001	<0.001	1	<0.001	<0.001
Laser	Plasma Spray	<0.001	0.001	<0.001	<0.001	<0.001	<0.001	<0.001	<0.001	<0.001
	HA-Blasted	<0.001	<0.001	<0.001	<0.001	<0.001	<0.001	<0.001	<0.001	<0.001
	Double Acid Etching	<0.001	<0.001	1	<0.001	<0.001	<0.001	<0.001	<0.001	<0.001
	HA-Blasted “New”	<0.001	0.025	<0.001	<0.001	0.002	<0.001	<0.001	1	0.022
HA-Blasted	Plasma Spray	<0.001	0.002	<0.001	<0.001	<0.001	<0.001	1	<0.001	<0.001
	Laser	<0.001	<0.001	<0.001	<0.001	<0.001	<0.001	<0.001	<0.001	<0.001
	Double Acid Etching	<0.001	<0.001	<0.001	<0.001	<0.001	<0.001	<0.001	0.218	1
	HA-Blasted “New”	<0.001	<0.001	<0.001	<0.001	<0.001	<0.001	1	<0.001	<0.001
Double Acid Etching	Plasma Spray	<0.001	<0.001	<0.001	0.008	<0.001	<0.001	<0.001	<0.001	<0.001
	Laser	<0.001	<0.001	1	<0.001	<0.001	<0.001	<0.001	<0.001	<0.001
	HA-Blasted	<0.001	<0.001	<0.001	<0.001	<0.001	<0.001	<0.001	0.218	1
	HA-Blasted “New”	<0.001	<0.001	<0.001	1	<0.001	<0.001	<0.001	<0.001	<0.001
HA-Blasted “New”	Plasma Spray	<0.001	0.314	<0.001	0.004	<0.001	<0.001	1	<0.001	<0.001
	Laser	<0.001	0.025	<0.001	<0.001	0.002	<0.001	<0.001	1	0.022
	HA-Blasted	<0.001	<0.001	<0.001	<0.001	<0.001	<0.001	1	<0.001	<0.001
	Double Acid Etching	<0.001	<0.001	<0.001	1	<0.001	<0.001	<0.001	<0.001	<0.001

## Data Availability

The datasets generated and/or analyzed during the current study are available upon reasonable request from the corresponding author.

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
