# Peer review of "Assessing the Efficacy of Whole-Body Titanium Dental Implant Surface Modifications in Inducing Adhesion, Proliferation, and Osteogenesis in Human Adipose Tissue Stem Cells"

_jfb, 2022, doi:10.3390/jfb13040206_

Round 1
Reviewer 1 Report
Assessing the efficacy of whole-body titanium dental implant
surface modifications in inducing adhesion, proliferation, and
osteogenesis in human adipose tissue stem cells
Comments:
1. The whole text needs to be proofread in English because there are errors
2. I suggest stating the abbreviations with an explanation in a special part of the text of the article - either at the beginning or at the end
3. the introductory part lacks the indication of the hypothesis of the study, which is shown below - it is not clearly visible enough
4. In Part 2.4. it would be necessary to state the chemical compositions of the dental implants that were used in the study → please insert in Table 1 the nominal chemical composition, together with microalloying elements and other potential impurities
5. I suggest that the article also includes the results of chemical analysis for the prepared surfaces of the implants that were subjected to the study → addition to Figure 3! It would be even better to add EDX mapping of the elements present on the surface of the implants to each image, since only these elements, which form different phases, are in direct connection with the tissue - therefore, they are responsible for all processes of osteointegration. Only in this way is it possible to describe and explain osteointegration together with the degree of roughness of the implant surface.
6. Tables 4, 5 are too complicated, at the level of a professional report - they need to be reformatted in accordance with the level of perception of the content of the scientific article - brevity and distinctness of the presentation of scientific results is needed
7. In the discussion part of lines 385-388, the interpretation of the obtained results is too short, very weak and relies only on the roughness of the surface, but does not include the chemical and phase composition of the implant surface, the stability of this surface, etc. Therefore, in this part of the text of the discussion, it is necessary to strengthen and refine it as required in point 5 → since the statement in lines 401-402 of the discussion, among other things, also speaks about the composition.
8. The conclusions must be reformatted, as this part repeats the content from the summary - they are inadequate!

Author Response
Dear Reviewer 1,
We appreciate your taking the time to read our paper. In this resubmitted version, we worked hard to address all of the comments and suggestions, and the paper has been much improved as a consequence of the reviewers' remarks.
We organized this letter to make the review process easier by addressing reviewers' points (shown in red) in the order they were made.
- The whole text needs to be proofread in English because there are errors
Ans: We appreciate the reviewer pointing out the mistakes. The entire article has been proofread and should now be free of errors.
- I suggest stating the abbreviations with an explanation in a special part of the text of the article - either at the beginning or at the end
Ans: The abbreviations list has been added at the end of the paper as kindly suggested by reviewer 1 and sounds as follows (page 18 line 505-512) “Abbreviations: adipose tissue derived stem cells (ADSCs), hydroxyapatite (HA), collagen type I (Coll-I), osteonectin (Osn), alkaline phosphatase (ALP), osteocalcin (Osc), arithmetical mean of the roughness area (Sa), fetal bovine serum (FBS), epithelial growth factor (EGF), bland detaching solution (CTC), cluster of differentiation stem markers (CD), DMEM high glucose (HG), transforming growth factor-3 (TGFβ-3), scanning electron microscope (SEM), root mean square of the surface roughness (Rq), roughness skewness of the surface (Rsk), roughness kurtosis of the surface (Rku), mean arithmetic roughness average (Ra), HA blasted and bland acid etching (HA blasted) and extracellular matrix (ECM)”.
- The introductory part lacks the indication of the hypothesis of the study, which is shown below - it is not clearly visible enough
Ans: Thanks to the reviewer, We have now added a more detailed section describing the study hypothesis, which is given in (page 2 lines 62-66) as follows, “Surprisingly, despite being a more realistic and clinically suitable approach[1], little is known about the impact of whole-body implants on stem cells, including attraction or adhesion and, secondly, proliferative and differentiation potential. Interestingly, stem cells have recently been identified as a combinatorial tissue engineering strategy to improve titanium implant osseointegration in diabetic and osteoporotic animal models[1,2]”.
”
- In Part 2.4. it would be necessary to state the chemical compositions of the dental implants that were used in the study →please insert in Table 1 the nominal chemical composition, together with microalloying elements and other potential impurities.
Ans: We are grateful to the reviewer for her/his suggestion and the nominal chemical composition of all the dental implants has been added into the revised version, line 135-155 and in table 1.
|
Surface treatment |
Plasma-spray |
Laser |
HA-Blasted and Bland Acid Etching |
Double Acid Etching |
HA-Blasted “new” under patent |
|
Composition |
Grade 4th titanium |
Grade 4th titanium |
Ti6Al4V |
Grade 4th titanium |
Ti6Al4V |
|
Nominal chemical composition[3] |
Ti 99% Fe 0.3% O 0.4% C 0.1% N 0.05% H 0.015% Al 0.0% V 0.0% |
Ti 99% Fe 0.3% O 0.4% C 0.1% N 0.05% H 0.015% Al 0.0% V 0.0% |
Ti 90% Fe 0.25% O 0.2% max C 0.0% N 0.0% H 0.0% Al 6.4% V 4.12% |
Ti 99% Fe 0.3% O 0.4% C 0.1% N 0.05% H 0.015% Al 0.0% V 0.0% |
Ti 90% Fe 0.25% O 0.2% max C 0.0% N 0.0% H 0.0% Al 6.4% V 4.12% |
- I suggest that the article also includes the results of chemical analysis for the prepared surfaces of the implants that were subjected to the study →addition to Figure 3! It would be even better to add EDX mapping of the elements present on the surface of the implants to each image, since only these elements, which form different phases, are in direct connection with the tissue - therefore, they are responsible for all processes of osteointegration. Only in this way is it possible to describe and explain osteointegration together with the degree of roughness of the implant surface.
Ans: We agree with the reviewer that energy dispersive x-ray spectroscopy analysis might be useful in this study. However, due to titanium supply shortages or delays, we have not been able to acquire all of the new and pristine dental implants from the suppliers. We understand that this is a significant limitation of the research, and we tried to address it by discussing the importance of the chemical surface composition, and citing earlier papers, which should hopefully provide a possible answer to the reviewer's question. The text, (page 17 line 435-455), is reported as follows: “A large body of evidence asserts that many contaminants, either metallic or nonmetallic such as C, Mg, Fe, Al, Ca, P, Sr, and F, are introduced on the implant surface voluntarily (commercially pure titanium grades and titanium alloys) or, in spite of strict control, during the manufacturing process or handling[4–6]. We previously demonstrated that machined and laser micro-patterned treatments showed no traceable surface impurity or modification, whereas "sandblasting" introduced elemental traces of C, Fe, Al, and O[7], thus chemically impacting the surface, changing its composition, and influencing in either way tissue and cells’ activity[4–6]. Another consideration that has to be taken into account is that any approach for modifying surface roughness varies the surface chemistry finally altering protein adsorption/adhesion, i.e., Ca, Mg [8].
As a result, it is possible to hypothesize that the procedures used in HA-blasted and bland acid etching implants modified both the chemical[4–6] and topographical properties, such as roughness or particle number[4,9], ultimately favoring cell adhesion[8].
Following the initial adhesion, which reached a peak for the HA-blasted and bland acid etching, a non-significant proliferative phase occurred for all but one implant, namely the laser treated implants on day three and the double acid etching implants on day seven. This is in contrast to previous studies on titanium discs[10–12] and it might be due to the low cell density or it could be related to the whole-body surface via the presence of solubilized metallic ions with negative effects, which cause inflammation and cytotoxicity as their concentration rises[13,14]” and (page 18 lines 484-486) “It is also plausible to speculate that the processes utilized in HA-blasted implants changed the chemical composition of the surface, i.e., adding Al, Ca, P, Na, S, Si, and O ions[4–6], thus promoting the differentiation of the ADSCs”.
- Tables 4, 5 are too complicated, at the level of a professional report - they need to be reformatted in accordance with the level of perception of the content of the scientific article - brevity and distinctness of the presentation of scientific results is needed
Ans: Thanks to the reviewer for bringing this to our attention, tables 4 and 5 have been revised and may now be more clear and understandable.
Table 4:
|
Reference Implant |
Implant Type |
24 hours |
3 days |
7 days |
|
Plasma Spray |
Laser |
<0.001 |
0.01 |
<0.001 |
|
HA Blasted |
0.152 |
0.003 |
0.125 |
|
|
Double Acid Etching |
0.234 |
0.054 |
0.999 |
|
|
HA Blasted “New” |
0.197 |
<0.001 |
<0.001 |
|
|
Laser |
Plasma Spray |
<0.001 |
0.01 |
<0.001 |
|
HA Blasted |
<0.001 |
<0.001 |
0.153 |
|
|
Double Acid Etching |
<0.001 |
0.034 |
0.049 |
|
|
HA Blasted “New” |
<0.001 |
<0.001 |
999 |
|
|
HA Blasted |
Plasma Spray |
0.152 |
0.003 |
0.125 |
|
Laser |
<0.001 |
<0.001 |
0.153 |
|
|
Double Acid Etching |
<0.001 |
<0.001 |
0.075 |
|
|
HA Blasted “New” |
<0.001 |
<0.001 |
0.113 |
|
|
Double Acid Etching |
Plasma Spray |
0.234 |
0.054 |
0.999 |
|
Laser |
<0.001 |
0.034 |
0.049 |
|
|
HA Blasted |
<0.001 |
<0.001 |
0.075 |
|
|
HA Blasted “New” |
<0.001 |
<0.001 |
0.001 |
|
|
HA Blasted “New” |
Plasma Spray |
0.197 |
<0.001 |
<0.001 |
|
Laser |
<0.001 |
<0.001 |
0.999 |
|
|
HA Blasted |
<0.001 |
<0.001 |
0.113 |
|
|
Double Acid Etching |
<0.001 |
<0.001 |
0.001 |
Table 5:
|
|
|
Coll-I |
ALP |
Osn |
||||||
|
Reference Implant |
Implant Type |
24 hours |
3 days |
7 days |
24 hours |
3 days |
7 days |
24 hours |
3 days |
7 days |
|
Plasma Spray |
Laser |
<0.001 |
0.001 |
<0.001 |
<0.001 |
<0.001 |
<0.001 |
<0.001 |
<0.001 |
<0.001 |
|
HA Blasted |
<0.001 |
0.002 |
<0.001 |
<0.001 |
<0.001 |
<0.001 |
1 |
<0.001 |
<0.001 |
|
|
Double Acid Etching |
<0.001 |
<0.001 |
<0.001 |
0.008 |
<0.001 |
<0.001 |
<0.001 |
<0.001 |
<0.001 |
|
|
HA Blasted "New" |
<0.001 |
0.314 |
<0.001 |
0.004 |
<0.001 |
<0.001 |
1 |
<0.001 |
<0.001 |
|
|
Laser |
Plasma Spray |
<0.001 |
0.001 |
<0.001 |
<0.001 |
<0.001 |
<0.001 |
<0.001 |
<0.001 |
<0.001 |
|
HA Blasted |
<0.001 |
<0.001 |
<0.001 |
<0.001 |
<0.001 |
<0.001 |
<0.001 |
<0.001 |
<0.001 |
|
|
Double Acid Etching |
<0.001 |
<0.001 |
1 |
<0.001 |
<0.001 |
<0.001 |
<0.001 |
<0.001 |
<0.001 |
|
|
HA Blasted "New" |
<0.001 |
0.025 |
<0.001 |
<0.001 |
0.002 |
<0.001 |
<0.001 |
1 |
0.022 |
|
|
HA Blasted |
Plasma Spray |
<0.001 |
0.002 |
<0.001 |
<0.001 |
<0.001 |
<0.001 |
1 |
<0.001 |
<0.001 |
|
Laser |
<0.001 |
<0.001 |
<0.001 |
<0.001 |
<0.001 |
<0.001 |
<0.001 |
<0.001 |
<0.001 |
|
|
Double Acid Etching |
<0.001 |
<0.001 |
<0.001 |
<0.001 |
<0.001 |
<0.001 |
<0.001 |
0.218 |
1 |
|
|
HA Blasted "New" |
<0.001 |
<0.001 |
<0.001 |
<0.001 |
<0.001 |
<0.001 |
1 |
<0.001 |
<0.001 |
|
|
Double Acid Etching |
Plasma Spray |
<0.001 |
<0.001 |
<0.001 |
0.008 |
<0.001 |
<0.001 |
<0.001 |
<0.001 |
<0.001 |
|
Laser |
<0.001 |
<0.001 |
1 |
<0.001 |
<0.001 |
<0.001 |
<0.001 |
<0.001 |
<0.001 |
|
|
HA Blasted |
<0.001 |
<0.001 |
<0.001 |
<0.001 |
<0.001 |
<0.001 |
<0.001 |
0.218 |
1 |
|
|
HA Blasted "New" |
<0.001 |
<0.001 |
<0.001 |
1 |
<0.001 |
<0.001 |
<0.001 |
<0.001 |
<0.001 |
|
|
HA Blasted "New" |
Plasma Spray |
<0.001 |
0.314 |
<0.001 |
0.004 |
<0.001 |
<0.001 |
1 |
<0.001 |
<0.001 |
|
Laser |
<0.001 |
0.025 |
<0.001 |
<0.001 |
0.002 |
<0.001 |
<0.001 |
1 |
0.022 |
|
|
HA Blasted |
<0.001 |
<0.001 |
<0.001 |
<0.001 |
<0.001 |
<0.001 |
1 |
<0.001 |
<0.001 |
|
|
Double Acid Etching |
<0.001 |
<0.001 |
<0.001 |
1 |
<0.001 |
<0.001 |
<0.001 |
<0.001 |
<0.001 |
|
- In the discussion part of lines 385-388, the interpretation of the obtained results is too short, very weak and relies only on the roughness of the surface, but does not include the chemical and phase composition of the implant surface, the stability of this surface, etc. Therefore, in this part of the text of the discussion, it is necessary to strengthen and refine it as required in point 5 →since the statement in lines 401-402 of the discussion, among other things, also speaks about the composition.
Ans: As per question 5 of the reviewer, Ans: We agree with the reviewer that energy dispersive x-ray spectroscopy analysis might be useful in this study. However, due to titanium supply shortages or delays, we have not been able to acquire all of the new and pristine dental implants from the suppliers. We understand that this is a significant limitation of the research, and we tried to address it by discussing the importance of the chemical surface composition, and citing earlier papers, which should hopefully provide a possible answer to the reviewer's question. The text, (page 17 line 435-455), is reported as follows: “A large body of evidence asserts that many contaminants, either metallic or nonmetallic such as C, Mg, Fe, Al, Ca, P, Sr, and F, are introduced on the implant surface voluntarily (commercially pure titanium grades and titanium alloys) or, in spite of strict control, during the manufacturing process or handling[4–6]. We previously demonstrated that machined and laser micro-patterned treatments showed no traceable surface impurity or modification, whereas "sandblasting" introduced elemental traces of C, Fe, Al, and O[7], thus chemically impacting the surface, changing its composition, and influencing in either way tissue and cells’ activity[4–6]. Another consideration that has to be taken into account is that any approach for modifying surface roughness varies the surface chemistry finally altering protein adsorption/adhesion, i.e., Ca, Mg [8].
As a result, it is possible to hypothesize that the procedures used in HA-blasted and bland acid etching implants modified both the chemical[4–6] and topographical properties, such as roughness or particle number[4,9], ultimately favoring cell adhesion[8].
Following the initial adhesion, which reached a peak for the HA-blasted and bland acid etching, a non-significant proliferative phase occurred for all but one implant, namely the laser treated implants on day three and the double acid etching implants on day seven. This is in contrast to previous studies on titanium discs[10–12] and it might be due to the low cell density or it could be related to the whole-body surface via the presence of solubilized metallic ions with negative effects, which cause inflammation and cytotoxicity as their concentration rises[13,14]” and (page 18 lines 484-486) “It is also plausible to speculate that the processes utilized in HA-blasted implants changed the chemical composition of the surface, i.e., adding Al, Ca, P, Na, S, Si, and O ions[4–6], thus promoting the differentiation of the ADSCs”.
- The conclusions must be reformatted, as this part repeats the content from the summary - they are inadequate!
Ans: Thanks about the reviewer’s suggestion, the conclusion has been changed and reformatted to not repeat the summary and reported in (page 18 lines 488-503) “In summary, we discovered that the surface roughness and treatment composition of dental implants play an important role in the adhesion and differentiation of ADSCs grown on whole-body titanium implants but not in their proliferation.
Significant progress has been achieved in applying material engineering to the study and modulation of a variety of restorative and pathological disorders[1,2,15–20], and preclinical attempts have been made to combine material engineering with stem cells[1]. Based on our findings, we suggest that HA-blasted titanium implants combined with adherent ADSCs, preconditioned[15,21] to increase their in vivo potential, might be helpful in clinical settings for people with systemic conditions such as diabetes and osteoporosis[1]. In such a way, it may be feasible to promote quicker osseointegration and increase the patient's quick tissue regeneration and bone production, therefore improving rehabilitation.
However, different seeding concentrations, cytotoxicity tests, longer culture intervals, and pre-clinical studies are still needed to fully grasp the advantages of HA-blasted implants over other types, and therefore, determine the most efficient and suitable surface to be employed in conjunction with stem cells”.
References
- Duan, Y.; Ma, W.; Li, D.; Wang, T.; Liu, B. Enhanced Osseointegration of Titanium Implants in a Rat Model of Osteoporosis Using Multilayer Bone Mesenchymal Stem Cell Sheets. Exp Ther Med 2017, 14, 5717–5726, doi:10.3892/etm.2017.5303.
- Xu, B.; Zhang, J.; Brewer, E.; Tu, Q.; Yu, L.; Tang, J.; Krebsbach, P.; Wieland, M.; Chen, J. Osterix Enhances BMSC-Associated Osseointegration of Implants. J Dent Res 2009, 88, 1003–1007, doi:10.1177/0022034509346928.
- W. Nicholson, J. Titanium Alloys for Dental Implants: A Review. Prosthesis 2020, 2, 100–116, doi:10.3390/prosthesis2020011.
- Albrektsson, T.; Wennerberg, A. On Osseointegration in Relation to Implant Surfaces. Clin Implant Dent Relat Res 2019, 21 Suppl 1, 4–7, doi:10.1111/cid.12742.
- Nishiguchi, S.; Kato, H.; Neo, M.; Oka, M.; Kim, H.M.; Kokubo, T.; Nakamura, T. Alkali- and Heat-Treated Porous Titanium for Orthopedic Implants. J Biomed Mater Res 2001, 54, 198–208, doi:10.1002/1097-4636(200102)54:2<198::aid-jbm6>3.0.co;2-7.
- Dhaliwal, J.S.; David, S.R.N.; Zulhilmi, N.R.; Sodhi Dhaliwal, S.K.; Knights, J.; de Albuquerque Junior, R.F. Contamination of Titanium Dental Implants: A Narrative Review. SN Applied Sciences 2020, 2, 1011, doi:10.1007/s42452-020-2810-4.
- Faccioni, F.; Bevilacqua, L.; Porrelli, D.; Khoury, A.; Faccioni, P.; Turco, G.; Frassetto, A.; Maglione, M. Ultrasonic Instrument Effects on Different Implant Surfaces: Profilometry, Energy-Dispersive X-Ray Spectroscopy, and Microbiology In Vitro Study. Int J Oral Maxillofac Implants 2021, 36, 520–528, doi:10.11607/jomi.8140.
- Barberi, J.; Spriano, S. Titanium and Protein Adsorption: An Overview of Mechanisms and Effects of Surface Features. Materials (Basel) 2021, 14, doi:10.3390/ma14071590.
- Kulangara, K.; Yang, J.; Chellappan, M.; Yang, Y.; Leong, K.W. Nanotopography Alters Nuclear Protein Expression, Proliferation and Differentiation of Human Mesenchymal Stem/Stromal Cells. PLoS One 2014, 9, e114698, doi:10.1371/journal.pone.0114698.
- Guida, L.; Annunziata, M.; Rocci, A.; Contaldo, M.; Rullo, R.; Oliva, A. Biological Response of Human Bone Marrow Mesenchymal Stem Cells to Fluoride-Modified Titanium Surfaces. Clin Oral Implants Res 2010, 21, 1234–1241, doi:10.1111/j.1600-0501.2010.01929.x.
- Zanicotti, D.G.; Duncan, W.J.; Seymour, G.J.; Coates, D.E. Effect of Titanium Surfaces on the Osteogenic Differentiation of Human Adipose-Derived Stem Cells. Int J Oral Maxillofac Implants 2018, 33, e77–e87, doi:10.11607/jomi.5810.
- Annunziata, M.; Guida, L.; Perillo, L.; Aversa, R.; Passaro, I.; Oliva, A. Biological Response of Human Bone Marrow Stromal Cells to Sandblasted Titanium Nitride-Coated Implant Surfaces. J Mater Sci Mater Med 2008, 19, 3585–3591, doi:10.1007/s10856-008-3514-2.
- Zhang, J.; Cai, B.; Tan, P.; Wang, M.; Abotaleb, B.; Zhu, S.; Jiang, N. Promoting Osseointegration of Titanium Implants through Magnesium- and Strontium-Doped Hierarchically Structured Coating. Journal of Materials Research and Technology 2022, 16, 1547–1559, doi:https://doi.org/10.1016/j.jmrt.2021.12.097.
- Stricker, A.; Bergfeldt, T.; Fretwurst, T.; Addison, O.; Schmelzeisen, R.; Rothweiler, R.; Nelson, K.; Gross, C. Impurities in Commercial Titanium Dental Implants - A Mass and Optical Emission Spectrometry Elemental Analysis. Dent Mater 2022, 38, 1395–1403, doi:10.1016/j.dental.2022.06.028.
- Ferro, F.; Spelat, R.; Shaw, G.; Coleman, C.M.; Chen, X.Z.; Connolly, D.; Palamá, E.M.F.; Gentili, C.; Contessotto, P.; Murphy, M.J. Regenerative and Anti-Inflammatory Potential of Regularly Fed, Starved Cells and Extracellular Vesicles In Vivo. Cells 2022, 11, doi:10.3390/cells11172696.
- Spelat, R.; Ferro, F.; Contessotto, P.; Warren, N.J.; Marsico, G.; Armes, S.P.; Pandit, A. A Worm Gel-Based 3D Model to Elucidate the Paracrine Interaction between Multiple Myeloma and Mesenchymal Stem Cells. Mater Today Bio 2020, 5, 100040, doi:10.1016/j.mtbio.2019.100040.
- Contessotto, P.; Pandit, A. Therapies to Prevent Post-Infarction Remodelling: From Repair to Regeneration. Biomaterials 2021, 275, 120906, doi:10.1016/j.biomaterials.2021.120906.
- Mayerhofer, C.C.K.; Ueland, T.; Broch, K.; Vincent, R.P.; Cross, G.F.; Dahl, C.P.; Aukrust, P.; Gullestad, L.; Hov, J.R.; Trøseid, M. Increased Secondary/Primary Bile Acid Ratio in Chronic Heart Failure. J Card Fail 2017, 23, 666–671, doi:10.1016/j.cardfail.2017.06.007.
- Marsico, G.; Jin, C.; Abbah, S.A.; Brauchle, E.M.; Thomas, D.; Rebelo, A.L.; Orbanić, D.; Chantepie, S.; Contessotto, P.; Papy-Garcia, D.; et al. Elastin-like Hydrogel Stimulates Angiogenesis in a Severe Model of Critical Limb Ischemia (CLI): An Insight into the Glyco-Host Response. Biomaterials 2021, 269, 120641, doi:10.1016/j.biomaterials.2020.120641.
- Limongi, T.; Brigo, L.; Tirinato, L.; Pagliari, F.; Gandin, A.; Contessotto, P.; Giugni, A.; Brusatin, G. Three-Dimensionally Two-Photon Lithography Realized Vascular Grafts. Biomed Mater 2021, 16, doi:10.1088/1748-605X/abca4b.
- Ferro, F.; Spelat, R.; Shaw, G.; Duffy, N.; Islam, M.N.; O’Shea, P.M.; O’Toole, D.; Howard, L.; Murphy, J.M. Survival/Adaptation of Bone Marrow-Derived Mesenchymal Stem Cells After Long-Term Starvation Through Selective Processes. Stem Cells 2019, 37, 813–827, doi:10.1002/stem.2998.

Reviewer 2 Report
The Authors presented a well designed research on the effect of the surface modifications of titanium dental implants surface on the receiving tissue capacity of ostointegration. However, some relevant aspects related to the potential local deformation (micro_strains at micro level) that can be induced by the different surface morphologies should be considered in the introduction and discussed in the conclusions.
The influence of mechanical micro_strains on dental implants ostointegration has been deeply investigated in previous literature.
Author Response
Dear Reviewer 2,
First and foremost, we want to thank you all for taking the time to read our manuscript. We worked hard to address all of the comments and concerns, and the article has been considerably improved as a result of the reviewers' observations in this resubmitted version.
We arranged this letter to simplify the review process by addressing reviewers' comments point-by-point (shown in red) in the same order they were made.
Reviewer 2
- The Authors presented a well designed research on the effect of the surface modifications of titanium dental implants surface on the receiving tissue capacity of ostointegration. However, some relevant aspects related to the potential local deformation (micro_strains at micro level) that can be induced by the different surface morphologies should be considered in the introduction and discussed in the conclusions. The influence of mechanical micro_strains on dental implants ostointegration has been deeply investigated in previous literature.
Ans: We agree with the reviewer and now the influence of the micro-strains have been added to the updated text both in the introduction (page 2 lines 58-60) and reported as follows: “Surface roughness is also linked to favorable effects on load transmission via the distribution of well-tolerated micro-strains, 0.25–0.50µε[22–24], which also favor osteoblastic and progenitor cell differentiation[22,24,25]” and in the conclusions: (page 17 line 426-432) “The stress caused by mastication loads on a dental implant creates dynamic strains on the surrounding tissues, thus affecting the rehabilitation process[22,26,24]. In view of this, the surface particles and roughness guarantee a differential load transfer depending on their physical and chemical characteristics[22,26,24]. Therefore, the presence of a configuration with the highest roughness and reduced number of particles on the HA-blasted and bland acid etching implants may have a considerably better impact on adhesion and compressive work-load distribution during and after rehabilitation[4,9]” and (page 18 line 478-482) “In general, our findings suggest that the surface topography (similar roughness or particle number and distribution) of the HA-treated surfaces promotes ADSCs' differentiation and possibly more durable osseointegration due to better load dissipation through the surrounding bone during rehabilitation[4,9]”.
References
- Li, J.; Jansen, J.A.; Walboomers, X.F.; van den Beucken, J.J. Mechanical Aspects of Dental Implants and Osseointegration: A Narrative Review. J Mech Behav Biomed Mater 2020, 103, 103574, doi:10.1016/j.jmbbm.2019.103574.
- Wang, L.; Aghvami, M.; Brunski, J.; Helms, J. Biophysical Regulation of Osteotomy Healing: An Animal Study. Clin Implant Dent Relat Res 2017, 19, 590–599, doi:10.1111/cid.12499.
- Leucht, P.; Kim, J.-B.; Wazen, R.; Currey, J.A.; Nanci, A.; Brunski, J.B.; Helms, J.A. Effect of Mechanical Stimuli on Skeletal Regeneration around Implants. Bone 2007, 40, 919–930, doi:10.1016/j.bone.2006.10.027.
- Wang, L.; Wu, Y.; Perez, K.C.; Hyman, S.; Brunski, J.B.; Tulu, U.; Bao, C.; Salmon, B.; Helms, J.A. Effects of Condensation on Peri-Implant Bone Density and Remodeling. J Dent Res 2017, 96, 413–420, doi:10.1177/0022034516683932.
- Kassem, R.; Samara, A.; Biadsee, A.; Masarwa, S.; Mtanis, T.; Ormianer, Z. A Comparative Evaluation of the Strain Transmitted through Prostheses on Implants with Two Different Macro-Structures and Connection during Insertion and Loading Phase: An In Vitro Study. Materials (Basel) 2022, 15, doi:10.3390/ma15144954.
- Albrektsson, T.; Wennerberg, A. On Osseointegration in Relation to Implant Surfaces. Clin Implant Dent Relat Res 2019, 21 Suppl 1, 4–7, doi:10.1111/cid.12742.
- Kulangara, K.; Yang, J.; Chellappan, M.; Yang, Y.; Leong, K.W. Nanotopography Alters Nuclear Protein Expression, Proliferation and Differentiation of Human Mesenchymal Stem/Stromal Cells. PLoS One 2014, 9, e114698, doi:10.1371/journal.pone.0114698.

Reviewer 3 Report
Dear Authors,
The sentence "Following tooth loss, bone and the surrounding tissues have the ability to repair, regenerate and support dental implant placement" should include healthy patient conditions.
The words "therapies and rehabilitations" have a specific meaning. The authors should present the word "osseointegration" regarding its definition by Albertson and colleagues (1981). I suggest replacing "therapies" (line 35) with rehabilitation.
The materials and methods section should present the inclusion and exclusion criteria, replacing "healthy volunteers." The authors should reference the protocol of the cell culture (lines 68 – 88) and all technical procedures. The authors should delete the control images in figure 2. The legends of the figures should identify and clarify the images. The sentence" Results are expressed as mean ± SD of 3 independent experiments, with * indicating significance p ≤ 0.05 (ANOVA followed by Bonferroni's post hoc test)" in figures 2, 3, 4, 5, 6, and 7 should be placed in the core of the manuscript.
The discussion section - the sentence "This leads us to assume that the cell death observed on the surface of the implants is most likely driven by integrin-mediated pro-apoptotic signaling, which occurs throughout the culture period" should be justified in the literature. The authors should discuss the clinical impact of their study.
In the conclusion section, lines 420-423 should be placed in the discussion section.
Author Response
Dear Reviewer 3,
We appreciate your time spent reading our publication. We worked hard in this resubmitted version to address all of the observations and suggestions, and the article has been greatly improved as a result of the reviewers' comments.
We arranged this letter to facilitate the review process by responding to reviewers' comments (shown in red) in the order they were presented.
- The sentence "Following tooth loss, bone and the surrounding tissues have the ability to repair, regenerate and support dental implant placement" should include healthy patient conditions.
Ans: Thanks for the suggestion, as the reviewer pointed out the sentence now introduces also the word healthy patients conditions and is reported as follows in (page 1 lines 36-38) “Healthy people typically recover quickly from dental implant surgery by repairing and regenerating the surrounding tissues, however, people suffering from pathological conditions, such as osteoporosis or diabetes, experience delays[2]”.
The words "therapies and rehabilitations" have a specific meaning.
Ans: Thanks to the reviewer the word therapies has been change with the word rehabilitation giving to the sentence the correct meaning and sounds like that “The effectiveness of the rehabilitation process after implant placement is dependent on appropriate epithelial and bone growth, which allows the device to osseointegrate at the implantation site.” in (page 1 lines 39-41).
- The authors should present the word "osseointegration" regarding its definition by Albertson and colleagues (1981).
Ans: As suggested by the reviewer the term osseointegration has been presented citing “Albrektsson T, Brånemark PI, Hansson HA, Lindström J. Osseointegrated titanium implants. Requirements for ensuring a long-lasting, direct bone-to-implant anchorage in man. Acta Orthop Scand. 1981;52(2):155-70. doi: 10.3109/17453678108991776. PMID: 7246093” and reported as follows in (page 1 lines 41-42) “The term "osseointegration" refers to the “formation of a direct contact between the dental implant and the living bone”[27]”.
- I suggest replacing "therapies" (line 35) with rehabilitation.
Ans: Thanks to the reviewer, the word therapies has been changed to rehabilitation, giving the sentence the correct meaning and sound in (page 1 lines 39-41) “The effectiveness of the rehabilitation process after implant placement is dependent on appropriate epithelial and bone growth, which allows the device to osseointegrate at the implantation site.”.
- The materials and methods section should present the inclusion and exclusion criteria, replacing "healthy volunteers."
Ans: Thanks, the inclusion criteria has been added to the materials and methods: (page 2 lines 77-79), “According to the ethical committee of the University of Udine, lipoaspirates were collected from healthy donors over the age of 18 after receiving informed consent and adhering to all legal criteria for confidentiality and management of biological material”
- The authors should reference the protocol of the cell culture (lines 68 – 88) and all technical procedures.
Ans: The refererence of the protocol has been added as well as the culture medium which was modified from Ferro F et al 2011. In (page 2,3 lines 80-101) and reported as follows: “Lipoaspirates (15ml) were taken from the donors utilizing the tumescent technique of infusion of Klein's solution followed by recovery of the adipose tissue[28]. After that, the adipose tissue was centrifuged at 1,800xg for 15 minutes to eliminate the red blood cells. ADSCs were put in jocklik modified alpha-MEM medium (Gibco, USA) supplemented with 400U/ml collagenase type 2 (Wortington, USA) equal to two times the volume of lipoaspirates, and it was incubated for 15 minutes at 37°C, with gentle shaking every 2-3 minutes[28]. The solution was then centrifuged at 1,800xg for 10 minutes [[28]]. The supernatant was discarded, and the pellet was resuspended and filtered through a filter to select the cell fraction smaller than 70μm[28]. ADSCs were plated into 100-mm dishes (2x106 cells per dish) and cultured in proliferation medium, adapted from Ferro F. et al., composed of α-MEM with L-glutamine 2mmol/l, 10% fetal bovine serum (FBS), insulin 10mg/ml, dexamethasone 10-9 mol/l, ascorbic acid 100mM, EGF 10ng/ml, and gentamicin 50ng/ml (all from Sigma-Aldrich, USA)[28]. Colonies developed in primary culture and reached near confluency in approximately one week. ADSCs were maintained semiconfluent to prevent cell differentiation, and approximately 80% of the medium was replaced every 3 days.
The stem cells derived from adipose tissue were maintained in culture medium at 37°C, 5% CO2, and humidified atmosphere. To detach ADSCs a bland detaching solution, namely CTC (collagenase 20U/ml, trypsin 0.75mg/ml, 2% heat-inactivated dialyzed chicken serum (Gibco) in Ca and Mg Hank's Balanced Salt Solution (HBSS), was used. Cells were maintained in culture from passages 1 to 6 (P1 to P6) in 150mm culture plates and then were used in the experiments[28]”.
- The authors should delete the control images in figure 2.
Ans: Following reviewer’s advice the control samples have been deleted from figure2:
- The legends of the figures should identify and clarify the images.
Ans: Following the reviewer's recommendations, the legends of the figures have been revised, and each letter now defines the specific image with which it is associated, increasing the readability of the image sequence.
- The sentence" Results are expressed as mean ± SD of 3 independent experiments, with * indicating significance p ≤ 0.05 (ANOVA followed by Bonferroni's post hoc test)" in figures 2, 3, 4, 5, 6, and 7 should be placed in the core of the manuscript.
Ans: The statistical summary of each experiment has been added into the main text of the paper. Specifically: in page 6 lines 215-217, page 7 lines 230-231, page 7 lines 247-248, page 10 lines 280-281, page 11 lines 289-291, page 12 lines 316-317, page 14 lines 343-345, and page 15 lines 370-371.
- The discussion section - the sentence "This leads us to assume that the cell death observed on the surface of the implants is most likely driven by integrin-mediated pro-apoptotic signaling, which occurs throughout the culture period" should be justified in the literature.
Ans: The sentence in the discussion has changed in this updated version of the manuscript however its substitute sentence has been referenced as kindly suggested by the reviewer and reported (page 18 lines 463-466) as follows: “It is known that once cells adhere to biomedical materials, their interaction elicits profound responses within the cells that, depending on how they are perceived, might result in proliferation, differentiation, and therefore survival, or apoptosis, and thus cell death[9,29–34]” .
- The authors should discuss the clinical impact of their study.
Ans: Authors’ are grateful to the reviewer for the advice to introduce the potential clinical impact of the study. The clinical impact of the study has been added to the end of the conclusions as per reviewer’s advice and is reported as follows (page 18 lines 491-499): “Significant progress has been achieved in applying material engineering to the study and modulation of a variety of restorative and pathological disorders[1,2,15–20], and preclinical attempts have been made to combine material engineering with stem cells[1]. Based on our findings, we suggest that HA-blasted titanium implants combined with adherent ADSCs, preconditioned[15,21] to increase their in vivo potential, might be helpful in clinical settings for people with systemic conditions such as diabetes and osteoporosis[1]. In such a way, it may be feasible to promote quicker osseointegration and increase the patient's quick tissue regeneration and bone production, therefore improving rehabilitation”.
In the conclusion section, lines 420-423 should be placed in the discussion section.
Ans: Discussion has been rewritten and lines 420-423 has been moved to the discussion as per keen suggestion of the reviewer, the text is reported (page 18 line 477-486) as follows “Even though neither the HA-blasted and bland acid etching nor the HA-blasted "New" implants produced significantly more ALP, Osn, or Coll-I than the other treatments, only the HA-blasted and bland acid etching could induce the expression of the late osteoblastic marker (Osc). In general, our findings suggest that the surface topography (similar roughness or particle number and distribution) of the HA-treated surfaces promotes ADSCs' differentiation and possibly more durable osseointegration due to better load dissipation through the surrounding bone during rehabilitation[4,9]. It is also plausible to speculate that the processes utilized in HA-blasted implants changed the chemical composition of the surface, i.e., adding Al, Ca, P, Na, S, Si, and O ions[4–6], thus promoting the differentiation of the ADSCs”.
References
- Duan, Y.; Ma, W.; Li, D.; Wang, T.; Liu, B. Enhanced Osseointegration of Titanium Implants in a Rat Model of Osteoporosis Using Multilayer Bone Mesenchymal Stem Cell Sheets. Exp Ther Med 2017, 14, 5717–5726, doi:10.3892/etm.2017.5303.
- Albrektsson, T.; Brånemark, P.I.; Hansson, H.A.; Lindström, J. Osseointegrated Titanium Implants. Requirements for Ensuring a Long-Lasting, Direct Bone-to-Implant Anchorage in Man. Acta Orthop Scand 1981, 52, 155–170, doi:10.3109/17453678108991776.
- Ferro, F.; Spelat, R.; Falini, G.; Gallelli, A.; D’Aurizio, F.; Puppato, E.; Pandolfi, M.; Beltrami, A.P.; Cesselli, D.; Beltrami, C.A.; et al. Adipose Tissue-Derived Stem Cell in Vitro Differentiation in a Three-Dimensional Dental Bud Structure. Am J Pathol 2011, 178, 2299–2310, doi:10.1016/j.ajpath.2011.01.055.
- Kulangara, K.; Yang, J.; Chellappan, M.; Yang, Y.; Leong, K.W. Nanotopography Alters Nuclear Protein Expression, Proliferation and Differentiation of Human Mesenchymal Stem/Stromal Cells. PLoS One 2014, 9, e114698, doi:10.1371/journal.pone.0114698.
- Abagnale, G.; Steger, M.; Nguyen, V.H.; Hersch, N.; Sechi, A.; Joussen, S.; Denecke, B.; Merkel, R.; Hoffmann, B.; Dreser, A.; et al. Surface Topography Enhances Differentiation of Mesenchymal Stem Cells towards Osteogenic and Adipogenic Lineages. Biomaterials 2015, 61, 316–326, doi:10.1016/j.biomaterials.2015.05.030.
- Steger M, Abagnale G, Bremus-Köbberling E, Wagner W, Gillner A. Nanoscale Biofunctionalization of Polymer Surfaces by Laser Treatment for Controlled Cellular Differentiation, in Optically Induced. Nanostructures:Biomedical and Technical Applications.; K. Konig and A. Ostendorf, Editors.: Berlin, 2015;
- Feller, L.; Jadwat, Y.; Khammissa, R.A.G.; Meyerov, R.; Schechter, I.; Lemmer, J. Cellular Responses Evoked by Different Surface Characteristics of Intraosseous Titanium Implants. Biomed Res Int 2015, 2015, 171945, doi:10.1155/2015/171945.
- Geiger, B.; Bershadsky, A.; Pankov, R.; Yamada, K.M. Transmembrane Crosstalk between the Extracellular Matrix--Cytoskeleton Crosstalk. Nat Rev Mol Cell Biol 2001, 2, 793–805, doi:10.1038/35099066.
- Cukierman, E.; Pankov, R.; Yamada, K.M. Cell Interactions with Three-Dimensional Matrices. Curr Opin Cell Biol 2002, 14, 633–639, doi:10.1016/s0955-0674(02)00364-2.
- Wozniak, M.A.; Modzelewska, K.; Kwong, L.; Keely, P.J. Focal Adhesion Regulation of Cell Behavior. Biochim Biophys Acta 2004, 1692, 103–119, doi:10.1016/j.bbamcr.2004.04.007.
- Xu, B.; Zhang, J.; Brewer, E.; Tu, Q.; Yu, L.; Tang, J.; Krebsbach, P.; Wieland, M.; Chen, J. Osterix Enhances BMSC-Associated Osseointegration of Implants. J Dent Res 2009, 88, 1003–1007, doi:10.1177/0022034509346928.
- Ferro, F.; Spelat, R.; Shaw, G.; Coleman, C.M.; Chen, X.Z.; Connolly, D.; Palamá, E.M.F.; Gentili, C.; Contessotto, P.; Murphy, M.J. Regenerative and Anti-Inflammatory Potential of Regularly Fed, Starved Cells and Extracellular Vesicles In Vivo. Cells 2022, 11, doi:10.3390/cells11172696.
- Spelat, R.; Ferro, F.; Contessotto, P.; Warren, N.J.; Marsico, G.; Armes, S.P.; Pandit, A. A Worm Gel-Based 3D Model to Elucidate the Paracrine Interaction between Multiple Myeloma and Mesenchymal Stem Cells. Mater Today Bio 2020, 5, 100040, doi:10.1016/j.mtbio.2019.100040.
- Contessotto, P.; Pandit, A. Therapies to Prevent Post-Infarction Remodelling: From Repair to Regeneration. Biomaterials 2021, 275, 120906, doi:10.1016/j.biomaterials.2021.120906.
- Mayerhofer, C.C.K.; Ueland, T.; Broch, K.; Vincent, R.P.; Cross, G.F.; Dahl, C.P.; Aukrust, P.; Gullestad, L.; Hov, J.R.; Trøseid, M. Increased Secondary/Primary Bile Acid Ratio in Chronic Heart Failure. J Card Fail 2017, 23, 666–671, doi:10.1016/j.cardfail.2017.06.007.
- Marsico, G.; Jin, C.; Abbah, S.A.; Brauchle, E.M.; Thomas, D.; Rebelo, A.L.; Orbanić, D.; Chantepie, S.; Contessotto, P.; Papy-Garcia, D.; et al. Elastin-like Hydrogel Stimulates Angiogenesis in a Severe Model of Critical Limb Ischemia (CLI): An Insight into the Glyco-Host Response. Biomaterials 2021, 269, 120641, doi:10.1016/j.biomaterials.2020.120641.
- Limongi, T.; Brigo, L.; Tirinato, L.; Pagliari, F.; Gandin, A.; Contessotto, P.; Giugni, A.; Brusatin, G. Three-Dimensionally Two-Photon Lithography Realized Vascular Grafts. Biomed Mater 2021, 16, doi:10.1088/1748-605X/abca4b.
- Ferro, F.; Spelat, R.; Shaw, G.; Duffy, N.; Islam, M.N.; O’Shea, P.M.; O’Toole, D.; Howard, L.; Murphy, J.M. Survival/Adaptation of Bone Marrow-Derived Mesenchymal Stem Cells After Long-Term Starvation Through Selective Processes. Stem Cells 2019, 37, 813–827, doi:10.1002/stem.2998.
- Albrektsson, T.; Wennerberg, A. On Osseointegration in Relation to Implant Surfaces. Clin Implant Dent Relat Res 2019, 21 Suppl 1, 4–7, doi:10.1111/cid.12742.
- Nishiguchi, S.; Kato, H.; Neo, M.; Oka, M.; Kim, H.M.; Kokubo, T.; Nakamura, T. Alkali- and Heat-Treated Porous Titanium for Orthopedic Implants. J Biomed Mater Res 2001, 54, 198–208, doi:10.1002/1097-4636(200102)54:2<198::aid-jbm6>3.0.co;2-7.
- Dhaliwal, J.S.; David, S.R.N.; Zulhilmi, N.R.; Sodhi Dhaliwal, S.K.; Knights, J.; de Albuquerque Junior, R.F. Contamination of Titanium Dental Implants: A Narrative Review. SN Applied Sciences 2020, 2, 1011, doi:10.1007/s42452-020-2810-4.

Reviewer 4 Report
General consideration:
This is an interesting piece of work considering that it aims to evaluate whether the micro-geometry and surface characteristics of commercially available implants with roughness between 1 and 2μm affect the behavior of a population of ADSCs. However, to make the work original, researchers are encouraged to look at properties that are not often characterized rather than a simple comparison between different materials. Works analyzing implant topography and cell adhesion are not new in the literature. Please make it clear the novelty of your work.
Format and content:
· Abstract. Please rewrite the abstract, making it more transparent and with the results found in your research project.
· Authors are also encouraged to provide substantial information about the rationale for the study design in the Introduction. For example, why is it essential for the clinician to know the surface roughness? Why use ADSCs and not fibroblast in this study? How would this affect the performance of an implant? How is this study different from other studies already published?
· Important details are missing in the Materials and Methods, and Results sections. Use “Ra” for arithmetical roughness. What is the chemical composition of the surface? Only anatase is present? What would be more challenging for cell attachment: the topography or the surface composition?
· When writing the discussion, please ensure the findings are presented and discussed/supported by information available in the literature.
This reviewer would also recommend a thorough English review so that the article may be written with a more scientific vocabulary.
Author Response
Dear Reviewer 4,
We appreciate your time spent reading our publication. We worked hard in this resubmitted version to address all of the observations and suggestions, and the article has been greatly improved as a result of the reviewers' comments.
We arranged this letter to facilitate the review process by responding to reviewers' comments (shown in red) in the order they were presented.
- Abstract. Please rewrite the abstract, making it more transparent and with the results found in your research project.
Ans: The abstract was checked, modified, and now should be more transparent and representative of the results found and reported at (page 1 line 13-28); “Background: Although the influence of titanium implants' micro-surface properties on titanium discs has been extensively investigated, the research has not taken into consideration the effect of their whole-body, which may be considered for a possible combinatorial approach. Methods: Five titanium dental implants with similar moderate roughness and different surface textures were thoroughly characterized. Cell adhesion and proliferation were assessed after adipose tissue derived stem cells (ADSCs) were seeded on whole-body implants. The implants' inductive properties were assessed by evaluating osteoblastic gene expression. Results: Surface micro-topography was analyzed, showing that hydroxyapatite (HA) blasted and bland acid etching implants had the highest roughness and a lower number of surface particles. Cell adhesion was observed after 24 hours on all implants, with the highest score registered for the HA blasted and bland acid etching implants. Cell proliferation was observed only on laser treated and double acid etched surfaces. ADSCs expressed collagen type I, osteonectin, and alkaline phosphatase on all implant surfaces, with high levels on HA-treated surfaces, which also triggered osteocalcin expression on day seven. Conclusions: The findings of this study show that the morphology and treatment of whole titanium dental implants, primarily HA-treated and bland acid etching implants, impact the adherence and activity of ADSCs toward osteogenic differentiation in the absence of specific osteo-inductive signals”.
- Authors are also encouraged to provide substantial information about the rationale for the study design in the Introduction. For example, why is it essential for the clinician to know the surface roughness? Why use ADSCs and not fibroblast in this study? How would this affect the performance of an implant? How is this study different from other studies already published?
We thank the reviewer for his or her suggestions, and we attempted to address all existing questions and concerns with an updated version of the introduction, which is reported on (page 1,2 lines 34-74) “It is estimated that in 2019, the global dental implant market was worth around 4.6 billion USD, and it is predicted to increase at an annual progression rate of 9.0% from 2020 to 2027[1]. Healthy people typically recover quickly from dental implant surgery by repairing and regenerating the surrounding tissues, however, people suffering from pathological conditions, such as osteoporosis or diabetes, experience delays[2].
The effectiveness of the rehabilitation process after implant placement is dependent on appropriate epithelial and bone growth, which allows the device to osseointegrate at the implantation site. The term "osseointegration" refers to the “formation of a direct contact between the dental implant and the living bone”[3]. Fast and tight osseointegration, therefore excellent long-term stability, are mandatory dental implants. The most important factor responsible for the stability and duration of an oral implant is likely the macro and micro-topography of the implant. In fact, properties, such as shape, elasticity, roughness, chemical composition, electric charge, oxide type, and thickness, have been demonstrated to play significant roles[4–11]. Theoretically, the surface geometry, charge, and their chemical-physical modifications have to fulfill four main tasks: (1) prevent the unspecific adsorption of denatured proteins at the interface between oral tissues and implants[12]; (2) attract differentiated or undifferentiated progenitor cells from the native tissue[13]; (3) induce native tissue or progenitor cell regeneration and differentiation[13]; (4) and guarantee an optimal load transfer to the bone[6,14]. Implants are mainly classified into four types based on surface roughness as determined by the arithmetical mean of the roughness area (Sa): rough (Sa>2.0 µm), moderately rough (Sa between 1.0-2.0 µm), minimally rough (Sa between 0.5-1.0 µm), smooth (Sa<0.5 µm)[15,16]. Many studies employing stem cells in vivo and in vitro to understand the mechanism of osseointegration[17–19] have found that implant titanium discs[20,21] with moderately rough surfaces enable better cell and bone connections than smoother or rougher surfaces[15,16,22–29]. Surface roughness is also linked to favorable effects on load transmission via the distribution of well-tolerated micro-strains, 0.25–0.50µε[6,30,31], which also favor osteoblastic and progenitor cell differentiation[6,31,32].
Surprisingly, despite being a more realistic and clinically suitable approach[2], little is known about the impact of whole-body implants on stem cells, including attraction or adhesion and, secondly, proliferative and differentiation potential. Interestingly, stem cell have recently been identified as a combinatorial tissue engineering strategy to improve titanium implant osseointegration in diabetic, and osteoporotic animal models[2,33].
Among stem cells, adipose tissue derived stem cells (ADSCs) are easily accessible and expandable in vitro, as well as capable of anti-inflammatory activity and differentiating along various lineages of paramount importance for implant dentistry, such as osteoblastic, ameloblastic, and odontoblastic lineages[34–37].
In detail, in the present research, we evaluate whether the micro-geometry and surface characteristics of five different whole-body commercially available titanium implants with roughness between 1 and 2μm affect the behavior of a population of ADSCs from the point of view of their adhesion, proliferation, and differentiation in vitro”.
- Important details are missing in the Materials and Methods, and Results sections. Use “Ra” for arithmetical roughness.
Sorry about the inaccuracy, citing Frias Martinez M.A. et al. “Roughness characteristics were examined in SEM photomicrographs using the SurfCharJ plugin (available for download at: http://imagej.nih.gov/ij/), which measures roughness parameters according to ISO 4287/2000: Ra (arithmetical mean deviation), Rq (root mean square deviation), Rku (kurtosis of the assessed profile), Rsk (skewness of the assessed profile)”. Anyway, we discovered that the surface roughness calculation plugin gives the same results as the SurfCharJ plugin. Furthermore, in this revised version, we changed the materials and methods, citing Martinez M.A. et al. and reporting the parameter definition in accordance with that paper throughout the manuscript. The text is reported on (page 4 lines 170-173) as follows: “Differences in roughness among the implants were assessed using SEM photomicrographs and the SurfCharJ plugin [13] to evaluate the following parameters: Rq (root mean square deviation), Rsk (skewness of the assessed profile), Rku (kurtosis of the assessed profile), and Ra (arithmetical mean deviation)”.
- What is the chemical composition of the surface? Only anatase is present?
Ans: We are grateful to the reviewer for her/his suggestion and the nominal chemical composition of all the dental implants has been added into the revised version, line 135-155 and in table 1.
|
Surface treatment |
Plasma-spray |
Laser |
HA-Blasted and Bland Acid Etching |
Double Acid Etching |
HA-Blasted “new” under patent |
|
Composition |
Grade 4th titanium |
Grade 4th titanium |
Ti6Al4V |
Grade 4th titanium |
Ti6Al4V |
|
Nominal chemical composition[38] |
Ti 99% Fe 0.3% O 0.4% C 0.1% N 0.05% H 0.015% Al 0.0% V 0.0% |
Ti 99% Fe 0.3% O 0.4% C 0.1% N 0.05% H 0.015% Al 0.0% V 0.0% |
Ti 90% Fe 0.25% O 0.2% max C 0.0% N 0.0% H 0.0% Al 6.4% V 4.12% |
Ti 99% Fe 0.3% O 0.4% C 0.1% N 0.05% H 0.015% Al 0.0% V 0.0% |
Ti 90% Fe 0.25% O 0.2% max C 0.0% N 0.0% H 0.0% Al 6.4% V 4.12% |
- What would be more challenging for cell attachment: the topography or the surface composition?
Thank you. We addressed the current question by revising and updating the discussion and conclusions reported on page 17, 18 (lines 409-503), “In simple terms, the effectiveness of dental implants depends on the creation of a barrier capable of both sealing the underlying osseous structures and integrating the body of the implant. It is well-known that the tissues interacting with the implant surface are essentially three. The first, closely bound to the implant, is poorly cellularized, and its ECM is composed of large and dense bundles of thick Coll-I fibers that contribute to the mechanical resistance and stability of the implants[39,40]. The second is relatively rich in fibroblasts with a large number of secretory components and is structurally made of Coll-I fibers that are heavily associated with collagen type III [39,40]. The third is the bone, which is mostly made up of inorganic mineral material HA with interspersed extracellular proteins, such as Coll-I, Osc, Osn, osteopontin, ALP, and a few scattered osteocytes[39,40].
In this study, we used whole titanium dental implants with moderately rough surfaces instead of titanium disks[17,20,23,41,42] and ADSCs, chosen because of their ease of separation as well as stemness properties[34], to deliver a more realistic approach and to assess the feasibility of a hypothetical combinatorial strategy.
The results revealed that, whereas the topological properties of the implant surfaces under consideration were similar[17,19,25,43], there were substantial differences in roughness, with the HA-blasted and bland acid etching exhibiting the highest value. Furthermore, while the HA-blasted and bland acid etching implants had the fewest surface particles, they also had the lowest density with respect to the other implants.
The stress caused by mastication loads on a dental implant creates dynamic strains on the surrounding tissues, thus affecting the rehabilitation process[6,14,31]. In view of this, the surface particles and roughness guarantee a differential load transfer depending on their physical and chemical characteristics[6,14,31]. Therefore, the presence of a configuration with the highest roughness and reduced number of particles on the HA-blasted and bland acid etching implants may have a considerably better impact on adhesion and load transfer during rehabilitation[19,23].
A large body of evidence asserts that many contaminants, either metallic or nonmetallic such as C, Mg, Fe, Al, Ca, P, Sr, and F, are introduced on the implant surface voluntarily (commercially pure titanium grades and titanium alloys) or, in spite of strict control, during the manufacturing process or handling[19,44,45]. We previously demonstrated that machined and laser micro-patterned treatments showed no traceable surface impurity or modification, whereas "sandblasting" introduced elemental traces of C, Fe, Al, and O[46], thus chemically impacting the surface, changing its composition, and influencing in either way tissue and cells’ activity[19,44,45]. Another factor that has to be taken into account is that any approach for modifying surface roughness varies the surface chemistry finally altering protein adsorption/adhesion, i.e., Ca, Mg[12].
As a result, it is possible to hypothesize that the procedures used in HA-blasted and bland acid etching implants modified both the chemical[19,44,45], and topographical properties, such as roughness or particle number[19,23], ultimately favoring cell adhesion[12].
Following the initial adhesion, which reached a peak for the HA-blasted and bland acid etching, a non-significant proliferative phase occurred for all but one implant, namely the laser treated implants on day three and the double acid etching implants on day seven. This is in contrast to previous studies on titanium discs[7,18,20] and it might be due to the low cell density or it could be related to the whole-body surface via the presence of solubilized metallic ions with negative effects, which cause inflammation and cytotoxicity as their concentration rises[47,48].
On the opposite, the findings suggest that the surface treatment and topography of the whole-body implants (roughness[18,26,27], surface particles density and characteristics[32], or potentially chemical composition[19,45,48]) have an active impact on ADSCs osteoblastic markers synthesis. In support, many studies employing stem cells in vivo and in vitro have found that dental implants[17–21]with moderately rough surfaces allow better osseointegration[15–29] and reduced marginal bone loss[24,49] than smoother or rougher surfaces.
It is known that once cells adhere to biomedical materials, their interaction elicits profound responses within the cells that, depending on how they are perceived, might result in proliferation, differentiation, and therefore survival, or apoptosis, and thus cell death[23,26,27,50–53].
The contact/interaction between the actin cytoskeleton and focal adhesion proteins with the implant surface has been identified as the critical regulator of the integration process[50]. Indeed, the integrin-mediated interaction between the extracellular matrix proteins such as fibronectin, vitronectin, osteonectin, and collagen-I and the implant surface has been documented to regulate cell adhesion, differentiation, and survival, via programmed cell death (apoptosis)[23,26,27,50–53]. Accordingly, we assume that following the adhesion, the cytoskeleton of the ADSCs underwent considerable reorganization, resulting in the intracellular signal transmission to the cytoplasm and nucleus[23,50]. The signals, in turn, triggered a variable degree of differentiation, coincident with the expression of increased levels of ALP, Osn, and Coll-I, starting on day one and lasting until day seven. Even though neither the HA-blasted and bland acid etching nor the HA-blasted "New" implants produced significantly more ALP, Osn, or Coll-I than the other treatments, only the HA-blasted and bland acid etching could induce the expression of the late osteoblastic marker (Osc). In general, our findings suggest that the surface topography (similar roughness or particle number and distribution) of the HA-treated surfaces promotes ADSCs' differentiation and possibly more durable osseointegration due to better load distribution through the surrounding bone during rehabilitation[19,23]. It is also plausible to speculate that the processes utilized in HA-blasted implants changed the chemical composition of the surface, i.e. adding Ca, P, Na, S, Si, and O ions[19,44,45], thus promoting the differentiation of the ADSCs. In summary, we discovered that the surface roughness and treatment composition of dental implants play an important role in the adhesion and differentiation of ADSCs grown on whole-body titanium implants, but not in their proliferation.
Significant progress has been achieved in applying material engineering to the study and modulation of a variety of restorative and pathological disorders[2,54–60], and preclinical attempts have been made to combine material engineering with stem cells[2]. Based on our findings, we suggest that HA-blasted titanium implants combined with adherent ADSCs, preconditioned[55,61] to increase their in vivo potential, might be helpful in clinical settings for people with systemic conditions, such as diabetes and osteoporosis[2]. In such a way, it may be feasible to promote quicker osseointegration and increase the patient's quick tissue regeneration and bone production, therefore improving rehabilitation.
However, different seeding concentrations, cytotoxicity tests, longer culture intervals, and pre-clinical studies are still needed to fully grasp the advantages of HA-blasted implants over other types, and therefore determine the most efficient and suitable surface to be employed in conjunction with stem cells”.
- When writing the discussion, please ensure the findings are presented and discussed/supported by information available in the literature.
Ans: Thanks to the reviewer, the discussion and conclusions have been checked and now the presented findings are supported by the previous literature.
- This reviewer would also recommend a thorough English review so that the article may be written with a more scientific vocabulary.
Ans: We appreciate that the reviewer pointed out the mistakes. The entire article has been proofread and should be error-free, with a more scientific vocabulary.
References
- Web source Dental Implants Market Size, Share & Trends Analysis Report By Type (Titanium, Zirconium), By Region (North America, Europe, Asia Pacific, Latin America, MEA), And Segment Forecasts, 2020 – 2027. 2020.
- Duan, Y.; Ma, W.; Li, D.; Wang, T.; Liu, B. Enhanced Osseointegration of Titanium Implants in a Rat Model of Osteoporosis Using Multilayer Bone Mesenchymal Stem Cell Sheets. Exp Ther Med 2017, 14, 5717–5726, doi:10.3892/etm.2017.5303.
- Albrektsson, T.; Brånemark, P.I.; Hansson, H.A.; Lindström, J. Osseointegrated Titanium Implants. Requirements for Ensuring a Long-Lasting, Direct Bone-to-Implant Anchorage in Man. Acta Orthop Scand 1981, 52, 155–170, doi:10.3109/17453678108991776.
- Coelho, P.G.; Granjeiro, J.M.; Romanos, G.E.; Suzuki, M.; Silva, N.R.F.; Cardaropoli, G.; Thompson, V.P.; Lemons, J.E. Basic Research Methods and Current Trends of Dental Implant Surfaces. J Biomed Mater Res B Appl Biomater 2009, 88, 579–596, doi:10.1002/jbm.b.31264.
- Insua, A.; Monje, A.; Wang, H.-L.; Miron, R.J. Basis of Bone Metabolism around Dental Implants during Osseointegration and Peri-Implant Bone Loss. J Biomed Mater Res A 2017, 105, 2075–2089, doi:10.1002/jbm.a.36060.
- Li, J.; Jansen, J.A.; Walboomers, X.F.; van den Beucken, J.J. Mechanical Aspects of Dental Implants and Osseointegration: A Narrative Review. J Mech Behav Biomed Mater 2020, 103, 103574, doi:10.1016/j.jmbbm.2019.103574.
- Guida, L.; Annunziata, M.; Rocci, A.; Contaldo, M.; Rullo, R.; Oliva, A. Biological Response of Human Bone Marrow Mesenchymal Stem Cells to Fluoride-Modified Titanium Surfaces. Clin Oral Implants Res 2010, 21, 1234–1241, doi:10.1111/j.1600-0501.2010.01929.x.
- Qahash, M.; Susin, C.; Polimeni, G.; Hall, J.; Wikesjö, U.M.E. Bone Healing Dynamics at Buccal Peri-Implant Sites. Clin Oral Implants Res 2008, 19, 166–172, doi:10.1111/j.1600-0501.2007.01428.x.
- Ripamonti, U.; Roden, L.C.; Ferretti, C.; Klar, R.M. Biomimetic Matrices Self-Initiating the Induction of Bone Formation. J Craniofac Surg 2011, 22, 1859–1870, doi:10.1097/SCS.0b013e31822e83fe.
- Löberg, J.; Gretzer, C.; Mattisson, I.; Ahlberg, E. Electronic Properties of Anodized TiO2 Electrodes and the Effect on in Vitro Response. J Biomed Mater Res B Appl Biomater 2014, 102, 826–839, doi:10.1002/jbm.b.33065.
- Vandrovcova, M.; Tolde, Z.; Vanek, P.; Nehasil, V.; Doubková, M.; Trávníčková, M.; Drahokoupil, J.; Buixaderas, E.; Borodavka, F.; Novakova, J.; et al. Beta-Titanium Alloy Covered by Ferroelectric Coating–Physicochemical Properties and Human Osteoblast-Like Cell Response. Coatings 2021, 11, doi:10.3390/coatings11020210.
- Barberi, J.; Spriano, S. Titanium and Protein Adsorption: An Overview of Mechanisms and Effects of Surface Features. Materials (Basel) 2021, 14, doi:10.3390/ma14071590.
- Martinez, M.A.F.; Balderrama, Í. de F.; Karam, P.S.B.H.; de Oliveira, R.C.; de Oliveira, F.A.; Grandini, C.R.; Vicente, F.B.; Stavropoulos, A.; Zangrando, M.S.R.; Sant’Ana, A.C.P. Surface Roughness of Titanium Disks Influences the Adhesion, Proliferation and Differentiation of Osteogenic Properties Derived from Human. Int J Implant Dent 2020, 6, 46, doi:10.1186/s40729-020-00243-5.
- Kassem, R.; Samara, A.; Biadsee, A.; Masarwa, S.; Mtanis, T.; Ormianer, Z. A Comparative Evaluation of the Strain Transmitted through Prostheses on Implants with Two Different Macro-Structures and Connection during Insertion and Loading Phase: An In Vitro Study. Materials (Basel) 2022, 15, doi:10.3390/ma15144954.
- Albrektsson, T.; Wennerberg, A. Oral Implant Surfaces: Part 1--Review Focusing on Topographic and Chemical Properties of Different Surfaces and in Vivo Responses to Them. Int J Prosthodont 2004, 17, 536–543.
- Albrektsson, T.; Wennerberg, A. Oral Implant Surfaces: Part 2--Review Focusing on Clinical Knowledge of Different Surfaces. Int J Prosthodont 2004, 17, 544–564.
- Cipriano, A.F.; De Howitt, N.; Gott, S.C.; Miller, C.; Rao, M.P.; Liu, H. Bone Marrow Stromal Cell Adhesion and Morphology on Micro- and Sub-Micropatterned Titanium. J Biomed Nanotechnol 2014, 10, 660–668, doi:10.1166/jbn.2014.1760.
- Zanicotti, D.G.; Duncan, W.J.; Seymour, G.J.; Coates, D.E. Effect of Titanium Surfaces on the Osteogenic Differentiation of Human Adipose-Derived Stem Cells. Int J Oral Maxillofac Implants 2018, 33, e77–e87, doi:10.11607/jomi.5810.
- Albrektsson, T.; Wennerberg, A. On Osseointegration in Relation to Implant Surfaces. Clin Implant Dent Relat Res 2019, 21 Suppl 1, 4–7, doi:10.1111/cid.12742.
- Annunziata, M.; Guida, L.; Perillo, L.; Aversa, R.; Passaro, I.; Oliva, A. Biological Response of Human Bone Marrow Stromal Cells to Sandblasted Titanium Nitride-Coated Implant Surfaces. J Mater Sci Mater Med 2008, 19, 3585–3591, doi:10.1007/s10856-008-3514-2.
- Giner, L.; Mercadé, M.; Torrent, S.; Punset, M.; Pérez, R.A.; Delgado, L.M.; Gil, F.J. Double Acid Etching Treatment of Dental Implants for Enhanced Biological Properties. J Appl Biomater Funct Mater 2018, 16, 83–89, doi:10.5301/jabfm.5000376.
- Rosa, M.B.; Albrektsson, T.; Francischone, C.E.; Schwartz Filho, H.O.; Wennerberg, A. The Influence of Surface Treatment on the Implant Roughness Pattern. J Appl Oral Sci 2012, 20, 550–555, doi:10.1590/s1678-77572012000500010.
- Kulangara, K.; Yang, J.; Chellappan, M.; Yang, Y.; Leong, K.W. Nanotopography Alters Nuclear Protein Expression, Proliferation and Differentiation of Human Mesenchymal Stem/Stromal Cells. PLoS One 2014, 9, e114698, doi:10.1371/journal.pone.0114698.
- Zheng, G.; Guan, B.; Hu, P.; Qi, X.; Wang, P.; Kong, Y.; Liu, Z.; Gao, P.; Li, R.; Zhang, X.; et al. Topographical Cues of Direct Metal Laser Sintering Titanium Surfaces Facilitate Osteogenic Differentiation of Bone Marrow Mesenchymal Stem Cells through Epigenetic Regulation. Cell Prolif 2018, 51, e12460, doi:10.1111/cpr.12460.
- Albouy, J.-P.; Abrahamsson, I.; Persson, L.G.; Berglundh, T. Implant Surface Characteristics Influence the Outcome of Treatment of Peri-Implantitis: An Experimental Study in Dogs. J Clin Periodontol 2011, 38, 58–64, doi:10.1111/j.1600-051X.2010.01631.x.
- Abagnale, G.; Steger, M.; Nguyen, V.H.; Hersch, N.; Sechi, A.; Joussen, S.; Denecke, B.; Merkel, R.; Hoffmann, B.; Dreser, A.; et al. Surface Topography Enhances Differentiation of Mesenchymal Stem Cells towards Osteogenic and Adipogenic Lineages. Biomaterials 2015, 61, 316–326, doi:10.1016/j.biomaterials.2015.05.030.
- Steger M, Abagnale G, Bremus-Köbberling E, Wagner W, Gillner A. Nanoscale Biofunctionalization of Polymer Surfaces by Laser Treatment for Controlled Cellular Differentiation, in Optically Induced. Nanostructures:Biomedical and Technical Applications.; K. Konig and A. Ostendorf, Editors.: Berlin, 2015;
- Svanborg, L.M.; Andersson, M.; Wennerberg, A. Surface Characterization of Commercial Oral Implants on the Nanometer Level. J Biomed Mater Res B Appl Biomater 2010, 92, 462–469, doi:10.1002/jbm.b.31538.
- Wennerberg, A.; Ide-Ektessabi, A.; Hatkamata, S.; Sawase, T.; Johansson, C.; Albrektsson, T.; Martinelli, A.; Södervall, U.; Odelius, H. Titanium Release from Implants Prepared with Different Surface Roughness. Clin Oral Implants Res 2004, 15, 505–512, doi:10.1111/j.1600-0501.2004.01053.x.
- Wang, L.; Aghvami, M.; Brunski, J.; Helms, J. Biophysical Regulation of Osteotomy Healing: An Animal Study. Clin Implant Dent Relat Res 2017, 19, 590–599, doi:10.1111/cid.12499.
- Leucht, P.; Kim, J.-B.; Wazen, R.; Currey, J.A.; Nanci, A.; Brunski, J.B.; Helms, J.A. Effect of Mechanical Stimuli on Skeletal Regeneration around Implants. Bone 2007, 40, 919–930, doi:10.1016/j.bone.2006.10.027.
- Wang, L.; Wu, Y.; Perez, K.C.; Hyman, S.; Brunski, J.B.; Tulu, U.; Bao, C.; Salmon, B.; Helms, J.A. Effects of Condensation on Peri-Implant Bone Density and Remodeling. J Dent Res 2017, 96, 413–420, doi:10.1177/0022034516683932.
- Kotsovilis, S.; Karoussis, I.K.; Fourmousis, I. A Comprehensive and Critical Review of Dental Implant Placement in Diabetic Animals and Patients. Clin Oral Implants Res 2006, 17, 587–599, doi:10.1111/j.1600-0501.2005.01245.x.
- Ferro, F.; Spelat, R.; Falini, G.; Gallelli, A.; D’Aurizio, F.; Puppato, E.; Pandolfi, M.; Beltrami, A.P.; Cesselli, D.; Beltrami, C.A.; et al. Adipose Tissue-Derived Stem Cell in Vitro Differentiation in a Three-Dimensional Dental Bud Structure. Am J Pathol 2011, 178, 2299–2310, doi:10.1016/j.ajpath.2011.01.055.
- Lee, J.A.; Parrett, B.M.; Conejero, J.A.; Laser, J.; Chen, J.; Kogon, A.J.; Nanda, D.; Grant, R.T.; Breitbart, A.S. Biological Alchemy: Engineering Bone and Fat from Fat-Derived Stem Cells. Ann Plast Surg 2003, 50, 610–617, doi:10.1097/01.SAP.0000069069.23266.35.
- Hattori, H.; Masuoka, K.; Sato, M.; Ishihara, M.; Asazuma, T.; Takase, B.; Kikuchi, M.; Nemoto, K.; Ishihara, M. Bone Formation Using Human Adipose Tissue-Derived Stromal Cells and a Biodegradable Scaffold. J Biomed Mater Res B Appl Biomater 2006, 76, 230–239, doi:10.1002/jbm.b.30357.
- Hicok, K.C.; Du Laney, T.V.; Zhou, Y.S.; Halvorsen, Y.-D.C.; Hitt, D.C.; Cooper, L.F.; Gimble, J.M. Human Adipose-Derived Adult Stem Cells Produce Osteoid in Vivo. Tissue Eng 2004, 10, 371–380, doi:10.1089/107632704323061735.
- W. Nicholson, J. Titanium Alloys for Dental Implants: A Review. Prosthesis 2020, 2, 100–116, doi:10.3390/prosthesis2020011.
- Zhang, B.; Li, J.; He, L.; Huang, H.; Weng, J. Bio-Surface Coated Titanium Scaffolds with Cancellous Bone-like Biomimetic Structure for Enhanced Bone Tissue Regeneration. Acta Biomater 2020, 114, 431–448, doi:10.1016/j.actbio.2020.07.024.
- Dohan Ehrenfest, D.M.; Piattelli, A.; Sammartino, G.; Wang, H.-L. New Biomaterials and Regenerative Medicine Strategies in Periodontology, Oral Surgery, Esthetic and Implant Dentistry 2018. Biomed Res Int 2019, 2019, 1363581, doi:10.1155/2019/1363581.
- Klein, M.O.; Bijelic, A.; Ziebart, T.; Koch, F.; Kämmerer, P.W.; Wieland, M.; Konerding, M.A.; Al-Nawas, B. Submicron Scale-Structured Hydrophilic Titanium Surfaces Promote Early Osteogenic Gene Response for Cell Adhesion and Cell Differentiation. Clin Implant Dent Relat Res 2013, 15, 166–175, doi:10.1111/j.1708-8208.2011.00339.x.
- Wennerberg, A.; Albrektsson, T. On Implant Surfaces: A Review of Current Knowledge and Opinions. Int J Oral Maxillofac Implants 2010, 25, 63–74.
- Albouy, J.-P.; Abrahamsson, I.; Berglundh, T. Spontaneous Progression of Experimental Peri-Implantitis at Implants with Different Surface Characteristics: An Experimental Study in Dogs. J Clin Periodontol 2012, 39, 182–187, doi:10.1111/j.1600-051X.2011.01820.x.
- Nishiguchi, S.; Kato, H.; Neo, M.; Oka, M.; Kim, H.M.; Kokubo, T.; Nakamura, T. Alkali- and Heat-Treated Porous Titanium for Orthopedic Implants. J Biomed Mater Res 2001, 54, 198–208, doi:10.1002/1097-4636(200102)54:2<198::aid-jbm6>3.0.co;2-7.
- Dhaliwal, J.S.; David, S.R.N.; Zulhilmi, N.R.; Sodhi Dhaliwal, S.K.; Knights, J.; de Albuquerque Junior, R.F. Contamination of Titanium Dental Implants: A Narrative Review. SN Applied Sciences 2020, 2, 1011, doi:10.1007/s42452-020-2810-4.
- Faccioni, F.; Bevilacqua, L.; Porrelli, D.; Khoury, A.; Faccioni, P.; Turco, G.; Frassetto, A.; Maglione, M. Ultrasonic Instrument Effects on Different Implant Surfaces: Profilometry, Energy-Dispersive X-Ray Spectroscopy, and Microbiology In Vitro Study. Int J Oral Maxillofac Implants 2021, 36, 520–528, doi:10.11607/jomi.8140.
- Zhang, J.; Cai, B.; Tan, P.; Wang, M.; Abotaleb, B.; Zhu, S.; Jiang, N. Promoting Osseointegration of Titanium Implants through Magnesium- and Strontium-Doped Hierarchically Structured Coating. Journal of Materials Research and Technology 2022, 16, 1547–1559, doi:https://doi.org/10.1016/j.jmrt.2021.12.097.
- Stricker, A.; Bergfeldt, T.; Fretwurst, T.; Addison, O.; Schmelzeisen, R.; Rothweiler, R.; Nelson, K.; Gross, C. Impurities in Commercial Titanium Dental Implants - A Mass and Optical Emission Spectrometry Elemental Analysis. Dent Mater 2022, 38, 1395–1403, doi:10.1016/j.dental.2022.06.028.
- Kämmerer, P.W.; Pabst, A.M.; Dau, M.; Staedt, H.; Al-Nawas, B.; Heller, M. Immobilization of BMP-2, BMP-7 and Alendronic Acid on Titanium Surfaces: Adhesion, Proliferation and Differentiation of Bone Marrow-Derived Stem Cells. J Biomed Mater Res A 2020, 108, 212–220, doi:10.1002/jbm.a.36805.
- Feller, L.; Jadwat, Y.; Khammissa, R.A.G.; Meyerov, R.; Schechter, I.; Lemmer, J. Cellular Responses Evoked by Different Surface Characteristics of Intraosseous Titanium Implants. Biomed Res Int 2015, 2015, 171945, doi:10.1155/2015/171945.
- Geiger, B.; Bershadsky, A.; Pankov, R.; Yamada, K.M. Transmembrane Crosstalk between the Extracellular Matrix--Cytoskeleton Crosstalk. Nat Rev Mol Cell Biol 2001, 2, 793–805, doi:10.1038/35099066.
- Cukierman, E.; Pankov, R.; Yamada, K.M. Cell Interactions with Three-Dimensional Matrices. Curr Opin Cell Biol 2002, 14, 633–639, doi:10.1016/s0955-0674(02)00364-2.
- Wozniak, M.A.; Modzelewska, K.; Kwong, L.; Keely, P.J. Focal Adhesion Regulation of Cell Behavior. Biochim Biophys Acta 2004, 1692, 103–119, doi:10.1016/j.bbamcr.2004.04.007.
- Xu, B.; Zhang, J.; Brewer, E.; Tu, Q.; Yu, L.; Tang, J.; Krebsbach, P.; Wieland, M.; Chen, J. Osterix Enhances BMSC-Associated Osseointegration of Implants. J Dent Res 2009, 88, 1003–1007, doi:10.1177/0022034509346928.
- Ferro, F.; Spelat, R.; Shaw, G.; Coleman, C.M.; Chen, X.Z.; Connolly, D.; Palamá, E.M.F.; Gentili, C.; Contessotto, P.; Murphy, M.J. Regenerative and Anti-Inflammatory Potential of Regularly Fed, Starved Cells and Extracellular Vesicles In Vivo. Cells 2022, 11, doi:10.3390/cells11172696.
- Spelat, R.; Ferro, F.; Contessotto, P.; Warren, N.J.; Marsico, G.; Armes, S.P.; Pandit, A. A Worm Gel-Based 3D Model to Elucidate the Paracrine Interaction between Multiple Myeloma and Mesenchymal Stem Cells. Mater Today Bio 2020, 5, 100040, doi:10.1016/j.mtbio.2019.100040.
- Contessotto, P.; Pandit, A. Therapies to Prevent Post-Infarction Remodelling: From Repair to Regeneration. Biomaterials 2021, 275, 120906, doi:10.1016/j.biomaterials.2021.120906.
- Mayerhofer, C.C.K.; Ueland, T.; Broch, K.; Vincent, R.P.; Cross, G.F.; Dahl, C.P.; Aukrust, P.; Gullestad, L.; Hov, J.R.; Trøseid, M. Increased Secondary/Primary Bile Acid Ratio in Chronic Heart Failure. J Card Fail 2017, 23, 666–671, doi:10.1016/j.cardfail.2017.06.007.
- Marsico, G.; Jin, C.; Abbah, S.A.; Brauchle, E.M.; Thomas, D.; Rebelo, A.L.; Orbanić, D.; Chantepie, S.; Contessotto, P.; Papy-Garcia, D.; et al. Elastin-like Hydrogel Stimulates Angiogenesis in a Severe Model of Critical Limb Ischemia (CLI): An Insight into the Glyco-Host Response. Biomaterials 2021, 269, 120641, doi:10.1016/j.biomaterials.2020.120641.
- Limongi, T.; Brigo, L.; Tirinato, L.; Pagliari, F.; Gandin, A.; Contessotto, P.; Giugni, A.; Brusatin, G. Three-Dimensionally Two-Photon Lithography Realized Vascular Grafts. Biomed Mater 2021, 16, doi:10.1088/1748-605X/abca4b.
- Ferro, F.; Spelat, R.; Shaw, G.; Duffy, N.; Islam, M.N.; O’Shea, P.M.; O’Toole, D.; Howard, L.; Murphy, J.M. Survival/Adaptation of Bone Marrow-Derived Mesenchymal Stem Cells After Long-Term Starvation Through Selective Processes. Stem Cells 2019, 37, 813–827, doi:10.1002/stem.2998.

Round 2
Reviewer 1 Report
Accept in present form
Reviewer 2 Report
The paper has been correctly revised including discussion and related references on the infuence of micro local strains
Reviewer 4 Report
Congratulations on your hard work.